# Quantum Chaos in the Dynamics of Molecules

**DOI:** 10.3390/e25010063

**Published:** 2022-12-29

**Authors:** Kazuo Takatsuka

**Affiliations:** Fukui Institute for Fundamental Chemistry, Kyoto University, Kyoto 606-8103, Japan; kaztak@fukui.kyoto-u.ac.jp

**Keywords:** quantum chaos, electronic-state chaos, electron dynamics, nonadiabatic dynamics, chemical dynamics, semiclassical mechanics, quantum tunneling, stochasticity and determinicity

## Abstract

Quantum chaos is reviewed from the viewpoint of “what is molecule?”, particularly placing emphasis on their dynamics. Molecules are composed of heavy nuclei and light electrons, and thereby the very basic molecular theory due to Born and Oppenheimer gives a view that quantum electronic states provide potential functions working on nuclei, which in turn are often treated classically or semiclassically. Therefore, the classic study of chaos in molecular science began with those nuclear dynamics particularly about the vibrational energy randomization within a molecule. Statistical laws in probabilities and rates of chemical reactions even for small molecules of several atoms are among the chemical phenomena requiring the notion of chaos. Particularly the dynamics behind unimolecular decomposition are referred to as Intra-molecular Vibrational energy Redistribution (IVR). Semiclassical mechanics is also one of the main research fields of quantum chaos. We herein demonstrate chaos that appears only in semiclassical and full quantum dynamics. A fundamental phenomenon possibly giving birth to quantum chaos is “bifurcation and merging” of quantum wavepackets, rather than “stretching and folding” of the baker’s transformation and the horseshoe map as a geometrical foundation of classical chaos. Such wavepacket bifurcation and merging are indeed experimentally measurable as we showed before in the series of studies on real-time probing of nonadiabatic chemical reactions. After tracking these aspects of molecular chaos, we will explore quantum chaos found in nonadiabatic electron wavepacket dynamics, which emerges in the realm far beyond the Born-Oppenheimer paradigm. In this class of chaos, we propose a notion of Intra-molecular Nonadiabatic Electronic Energy Redistribution (INEER), which is a consequence of the chaotic fluxes of electrons and energy within a molecule.

## 1. Introductory Remarks: What Makes Molecules Special in Quantum Science
and Chaos

Molecules consist of heavy nuclei and light electrons, which give rise to quite interesting and characteristic properties. All of the living bodies and biological systems on the globe are made up with molecules and are subject to the beautiful chemical laws. In the biological activities most of those molecular dynamics and properties should appear to be *deterministic* in the molecular levels, whereas the general dynamics of molecules often work in *stochastic* and *statistical* manners. The origin of such statistical properties arising even from small molecular systems are widely ascribed to classical [1] and quantum [2,3,4,5,6,7,8,9,10,11,12] chaos. This in turn suggests that one may find novel chaos in the dynamics of molecules.

Light electrons in a molecule move very fast and yet are accompanied with long wavelength in their matter waves, while the nuclear motion is much slower, but the relevant wavelengths are relatively far shorter due to the heavy masses. Thus, the two intrinsic hierarchical structures emerge in time-scale and length-scale within the dynamics of a molecule. Consequently the general picture of a molecule are: (i) Nuclei move on mean-field potential functions that are generate by the electrons. (ii) The nuclear motion can bear classical and semiclassical nature due to the short wavelength, while electrons are utterly quantum. Therefore, in contrast to classical chaos of the nuclear motion, it is often claimed that there is no chaos in electronic states because of discreteness in the energy levels with large gaps among them. Physics of the mapping of classical chaos onto quantum systems is often referred to as quantum chaology [13]. As a matter of fact most of the chaos studies in molecular science have been focused on the vibrational spectroscopy and nonlinear mixing of the vibrational modes and the theoretical foundation of statistical laws in chemical reaction dynamics. All these are about the chaoticity arising from the nuclear motion alone.

However, the theoretical separation of the electronic and nuclear motions due to the fast and slow modes, which is known as the Born-Oppenheimer separation, can sometimes break down through large kinematic interaction between them, which cause a mixing of the relevant electronic states. This interaction is generally and vaguely referred to as nonadiabatic interaction [14,15,16,17,18,19,20]. This interaction is extremely important in molecular science, since the electronic states and the relevant molecular properties can change instantaneously and drastically when a molecule passes through those nonadiabatic interaction regions. Moreover, we will show that nonadiabatic mixing among many electronic states involved in the manifold of densely quasi-degenerate electronic states can result in strong chaos, which can manifest as a diffusion motion of electronic states in the Hilbert space. This article will hence discuss chaos in what we call nonadiabatic electron dynamics [20,21,22,23,24,25,26,27].

Chaos theory for Hamilton mechanics was initiated (resumed from the view point of Poincaré) around 1950s [28]. Many beautiful theoretical models, elegant mathematics of geometry along with sophisticated numerical studies on ordinary differential equations have been studied since [1]. As for molecular chaos, there are very many pieces of theoretical studies in the literature, yet, due to the complexity of molecular nature most of the theoretical developments and applications are limited to small dimensional systems, typically two-dimensional in the early days. On the other side, the present author and his group have long been studying the chemical dynamics ranging from nuclear classical and semiclassical dynamics (see ref. [29,30] and more cited there) to full quantum electron dynamics [25,26,27]. Such molecular studies on chemical dynamics often guide us to chaotic phenomena that appear only in molecules. Conversely, tracking the path of our chaos studies, along with some data added herewith, would give partly an idea about what the dynamics of molecule are. We therefore apologize in advance that this review article covers only a very biased aspect of chaos studies. We hence do not cover the topics of the main-stream theories and well-established materials for quantum chaos such as level statistics, the random matrix theory, chaos in external fields and dissipative systems, and so on. Extensive reviews about general quantum chaos theory [11,12] are now widely available in numerous monographs [2,3,4,5,6,7,8,9,10] and review articles published elsewhere. The special issue of this journal Entropy, “Quantum Chaos—Dedicated to Professor Giulio Casati on the Occasion of His 80th Birthday”, is certainly among the newest ones.

We begin this review with the very basic structure of molecular science in Section 2, along with the so-called Born-Oppenheimer approximation, which separates nuclear and electronic dynamics in molecules [31].

In Section 3, we briefly touch upon the aims and implications of chaos studies in molecular science, particularly for chemical dynamics. For this purpose only, we take a couple of examples from previous studies on classical chaos.

Section 4 resumes with the full quantum chaos using the modified Hénon-Heiles potential as a model system of two-dimensional vibrational motions. We track the quantum wavepackets running within the quasi-separatrix and construct some of the associated eigenfunctions. A class of dynamical tunneling that exists only in the quasi-separatrix will be presented. The long-time energy spectra and the dependence of the chaotic spectra on the magnitude of the Planck constant are briefly examined.

Multidimensional semiclassical mechanics is indispensable in the treatment of the dynamics of heavy nuclei in molecules. We review our developed semiclassics [29,30] in Section 5, which we call the method of Action Decomposed Function (ADF) [32,33]. With ADF we analyze the mechanism of quantization not only of forming the energy peaks but of nullification of the off-resonance components. As a consequence of our theoretical and numerical studies, we proceed to the idea of amplitude free quantization without use of the annoying amplitude factors such as the van Vleck determinant [34] and those inherent to the semiclassical kernels (Greens functions) [35,36,37]. Semiclassical tunneling is also touched upon.

Then we study the characteristic molecular science, in which the nuclear dynamics kinematically couples with the electronic states, where the electronic-nuclear separation is broken down. In Section 6, we first concentrate on the nonadiabatic nuclear dynamics, which are subject to coupled equations of motion on multiple potential energy surfaces. The electronic states are still represented in time-independent wavefunctions. We show that the phenomenon of “bifurcation and merging (remixing) of quantum wavepackets”, a fundamental mechanism of quantum chaos, is indeed observable experimentally.

The final part of this review is devoted to the nonadiabatic electron dynamics in Section 7. Electronic wavepacket description is particularly useful in the stage where the adiabatic electronic states behind are embedded in a quasi-degenerate manifold and thereby nonadiabatic mixing among them is very intense and continual. Those strong and enduring electronic state mixing result in quantum chaos for the molecular electronic states.

This review concludes with some remarks in Section 8, stressing that the theory of quantum chaos and philosophy behind is indeed fundamental and useful also in practical science like chemistry.

## 2. Brief Introduction to the Theoretical Framework of Molecules:
Born-Oppenheimer Approximation to Separate Electronic and Nuclear
Motions

We first outline briefly the very basic frame of quantum description of molecules suggesting how chaos studies of molecules proceed.

### 2.1. The Born-Oppenheimer (BO) Approximation

The nonrelativistic molecular Hamiltonian is given as
(1)$$\begin{array}{cc} H(r,R) & =T_{N}+H^{el}\left(r;R\right)\\ & =\frac{1}{2}\sum_{k}\frac{{\hat{P}}_{k}^{2}}{M_{k}}+H^{\left(el\right)}\left(r;R\right) \end{array}$$
and
(2)$$i\hbar \frac{\partial }{\partial t}\Psi \left(r,R,t\right)=H\left(r,R\right)$$is the electronic Hamiltonian defined as
(3)$$\begin{array}{cc} H^{\left(el\right)}\left(r;R\right) & =T_{e}+V_{c}\left(r;R\right)\\ & =\frac{1}{2m}\sum_{j}{\hat{p}}_{j}^{2}+V_{c}\left(r;R\right) \end{array}$$
with r and R representing the electronic and nuclear coordinates, respectively, while ${\hat{p}}_{j}$ and ${\hat{P}}_{k}$ denote the operators for their conjugate momenta of the *j*th and *k*th component of r (specifically written as $r_{j}$) and R ($R_{k}$), respectively. We here consider only the Coulombic interactions $V_{c}\left(r;R\right)$ among electrons and nuclei, that is
(4)$$V_{c}\left(r;R\right)=\sum_{a<b}\frac{e^{2}}{\left|r_{a}-r_{b}\right|}-\sum_{a}\sum_{A}\frac{Z_{A}e^{2}}{\left|r_{a}-R_{A}\right|}+\sum_{A<B}\frac{Z_{A}Z_{B}e^{2}}{\left|R_{A}-R_{B}\right|},$$
where $r_{a}$ and $R_{A}$ are the *a*th electron and *A*th nuclei, respectively, and $Z_{A}$ indicate the nuclear charge on the *A*th nuclei.

Due to the great difference in the time scales between electrons and nuclei, their motion is assumed to be separated at each time, as the so-called the Born-Oppenheimer (BO) approximation represents the total wavefunction Ψ(r,R,t) to be
(5)$$\Psi \left(r,R,t\right)≃\chi_{I}\left(R,t\right)\Phi_{I}\left(r;R\right),$$
where $\Phi_{I}\left(r;R\right)$ and $\chi_{I}\left(R,t\right)$ are the electronic and nuclear wavefunctions, respectively. Note that electrons are assumed to move so fast that it can adjust themselves as a stationary wave at any nuclear position R, and thereby the electronic wavefunction $\Phi_{I}\left(r;R\right)$ is assumed to be free of time coordinate *t*. This in turn requires that $\Phi_{I}\left(r;R\right)$ is to be attained at each R such that
(6)$$H^{el}\left(r;R\right)\Phi_{I}\left(r;R\right)=V_{I}\left(R\right)\Phi_{I}\left(r;R\right).$$

Notice that R in $\Phi_{I}\left(r;R\right)$ and in Equation (Equation 6) is thereby treated as a parameter. The electronic energy $V_{I}\left(R\right)$ as a function of nuclear coordinates serves as a potential energy function (I=1,2,…) for the nuclear wavefunction in such a manner that
(7)$$i\hbar \frac{\partial }{\partial t}\chi_{I}\left(R,t\right)=\left[\sum_{k}\frac{{\hat{P}}_{k}^{2}}{2M_{k}}\chi +V_{I}\left(R\right)\right]\chi_{I}\left(R,t\right).$$

The time independent figure $V_{I}\left(R\right)$ is often called the potential energy hyper-surface (PES) in the chemical literature [31].

The lowest-order correction to the Born–Oppenheimer energy is found to be in the order of
(8)$${\left(m/M\right)}^{6/4}.$$

This analysis [38] clearly demonstrates that the Born–Oppenheimer approximation is far better than would be expected from a simple mass ratio m/M. For example, if $m/M∼10^{-4}$, which is typical in molecular systems, the error term is ${\left(m/M\right)}^{1.5}∼10^{-6}$.

### 2.2. The Born-Huang Expansion

To approach the exact solution of Equation (Equation 2) from the BO approximation, we need to expand Ψ(r,R,t) in the stationary electronic basis functions $\left\{\Phi_{I}\left(r;R\right)\right\}$ at each nuclear position R as
(9)$$\Psi \left(r,R,t\right)=\sum_{I}\chi_{I}\left(R,t\right)\Phi_{I}\left(r;R\right),$$
where unknown time-dependent coefficients $\chi_{I}\left(R,t\right)$ are supposed to describe the *nuclear wavepackets*. This is referred to as the Born-Huang expansion [39]. The electronic basis functions $\left\{|\Phi_{I}\rangle \right\}$ are assumed to be orthonormal at each nuclear configuration as
(10)$$\langle \Phi_{I}\left(R\right)|\Phi_{J}\left(R\right)\rangle =\delta_{IJ}.$$

The nuclear wavefunctions are subject to the following coupled equations
(11)$$i\hbar \frac{\partial }{\partial t}\chi_{I}=\frac{1}{2}\sum_{k}\frac{{\hat{P}}_{k}^{2}}{M_{k}}\chi_{I}+\sum_{J}H_{IJ}^{el}\chi_{J}-i\hbar \sum_{k}\sum_{J}\frac{X_{IJ}^{k}}{M_{k}}{\hat{P}}_{k}\chi_{J}-\frac{\hbar^{2}}{2}\sum_{k}\sum_{J}\frac{Y_{IJ}^{k}}{M_{k}}\chi_{J},$$
where
(12)$$X_{IJ}^{k}=\langle \left.\Phi_{I}\right|\frac{\partial \Phi_{J}}{\partial R_{k}}\rangle ,\;\;\;\;Y_{IJ}^{k}=\langle \Phi_{I}|\frac{\partial^{2}\Phi_{J}}{\partial {R_{k}}^{2}}\rangle ,$$
and
(13)$$H_{IJ}^{el}\left(R\right)=\langle \Phi_{I}\left|H^{el}\right|\Phi_{J}\rangle .$$

Since $X_{IJ}^{k}$ and $Y_{IJ}^{k}$ serve as the mixing elements among the adiabatic electronic states, they are collectively called nonadiabatic interactions. Tough difficulties in an attempt to solve the coupled equations of motion (Equation 11) are due to the many dimensionality of R, the oscillatory nature (due to the short wavelength) of $\left\{\chi_{I}\right\}$, and many channels to be taken into account. Besides, some of $\left\{\chi_{I}\right\}$ can extend to asymptotic regions in reaction (scattering) dynamics.

Note that the above Born-Huang expansion can be performed with use of other basis functions that do not satisfy Equation (Equation 6). In such a case $H_{IJ}^{el}\left(R\right)$ in Equation (Equation 13), have significant amount of nonzero off-diagonal elements. $\left\{\Phi_{I}\left(r;R\right)\right\}$ satisfying Equation (Equation 6) are referred to as the *adiabatic* wavefunctions, while others are to *diabatic* wavefunctions collectively. There are many ways to determine the diabatic functions [16,17,18,20], a typical set being those that minimize $\left|X_{IJ}^{k}\right|$. We will use the adiabatic representation in what follow unless otherwise noted.

## 3. Implication of Chaos in Chemical Dynamics

### 3.1. Determinicity versus Stochasticity in Molecules

It was Arrhenius who established the well-known thermodynamical reaction rate *k* from initial molecules to products in the form
(14)$$\;\;k\propto \exp(-E_{a}/RT).$$

This formula is indeed revolutionary in that the *rate or speed* has been derived based on the Maxwell-Boltzmann distribution for *equilibrium states*. The physical insight of this rate expression was uncovered by Eyring [40,41,42]. He studied a potential energy surface for simple atomic exchange reaction
(15)$$A\;+\;\mathrm{BC}\;\to \;{\left[\mathrm{ABC}\right]}^{‡}\to \mathrm{AB}\;+\;C$$
where ${\left[\mathrm{ABC}\right]}^{‡}$ is a transient state, called the transition state, at the bottleneck area on the way of reaction. He assumed a statistical distribution for the vibrational modes transversal to the translational one and associated the partition functions with them. He thus clarified the quantum mechanical meaning of the barrier height $E_{a}$ along with the preexponential factors to *k* (Polanyi and Wigner also stressed the geometrical roles of the potential energy surface $V_{I}\left(R\right)$ of Equation (Equation 6)) [40,41,42]. It is now known that the transition state theory is approximately valid even in binary collisions in vacuum. Even a quantum mechanical extension is one of the important subjects [43,44]. On the other hand, rather simple reactions for small systems can now well be treated in full quantum mechanics in a completely deterministic manner. (I do never mean that the full quantum scattering calculation for chemical reactions is an easy task. See refs. [45,46] for examples.) Yet the fundamental question is aimed at the theoretical ground of the statistical assumption made in the transition state.

While being subject to statistical laws on one hand, the dynamics of molecular motions are generally deterministic on the other hand. Let us see such a rather intricate example. See Figure 1, which is an ab initio (first principle) dynamics of 5-methyltropolone (5MTR). Ushiyama found that the proton shift from on O${}_{2}$ in the panel (a) to the position next to O${}_{1}$ (panel (c)) induces the rotation of the methyl group -C${}_{8}$H${}_{a}$H${}_{b}$H${}_{c}\;$at the bottom. The reciprocal motion of proton turns to rotational motion of the methyl group through such a long distance. Careful inspection at the panel (a) and (c) shows that this determinicity is supported by the shift of double bonds (see ref. [47] for the electronic state mechanism). This example denies the simple minded view that a molecule can be always regarded as a set of points of nuclear point masses connected by “springs” among them. We therefore need a careful thought when we study the chaos of molecular vibrational motions.

Another example of statistical reaction rate theory was given by Marcus [48], which augmented the successors’ formalism due to Rice, Ramsperger, and Kassel, thereby now called the RRKM theory for unimolecular dissociation, which predicts the rate of slow unimolecular dissociation of a molecule energized high enough to surmount the potential barrier. This process is schematically represented as
(16)$${\left[\mathrm{MA}\right]}^{*}\to M\;+\;A,$$
where an energized molecule ${\left[\mathrm{MA}\right]}^{*}$ may have been produced by collision and/or photo-excitation, and so on. From the view point of quantum scattering theory, ${\left[\mathrm{MA}\right]}^{*}$ can be regarded as a core (temporally) excited resonance, and the unimolecular dissociation may be viewed as a half-collision process of the Feshbach resonance [49,50]. In case where those decaying vibrational energy levels are densely packed having high density of states, a state in the vibrational-state manifold wanders among many nonlinearly coupled vibrational states with a long life-time. It is intuitively clear that statistical prediction should work well in such a situation, since the corresponding microcanonical ensemble is expected to be highly mixing. The years of 1970s have been devoted to prove that the ergodicity and mixing in the relevant classical phase space of ${\left[\mathrm{MA}\right]}^{*}$ is indeed realized by classical chaos. Such stochastic energy redistribution within a molecule is now generally termed Intramolecular Vibrational energy Redistribution (IVR), which represents one of the most important concepts in chemical dynamics and will be briefly touched upon below. In short, statistical mechanics works surprisingly well even in such small molecules composed only of several atoms, not of the Avogadro number. Those dynamics deviating from the RRKM theory are now placed under special focus for their possible peculiar phase-space structures, which are often termed as non-RRKM dynamics.

The key quantity that brings about stochastic and statistical nature into small molecular dynamical systems is the density of state (DOS), not the number of particles (atoms and molecules). In case where the DOS is low, the quantum dynamics should be subject to the Schrödinger equation and naturally deterministic. The higher is DOS, the harder become the relevant coupled equations to solve, and statistical treatments turn out to be more feasible. Due to the heavy masses of nuclei, the vibrational and rotational degrees of freedom of a molecule tend to have very high DOS, which can readily amount to the order 10${}^{10}$ eV${}^{-1}$ even for several atomic systems. This is not as large as the Avogadro number but sufficiently large to allow for statistical treatment. On the other hand, the DOS for the typical low-lying *electronic states* is less than 1.0 eV${}^{-1}$, and quantum mechanics is almost only one faithful approach. For this reason, the theory of stationary molecular electronic states is widely called quantum chemistry. For these low DOS and discrete states, it is a natural assertion that there is no chaos. Yet, in the end of this review, we will penetrate the *electronic state manifold* of densely quasi-degenerate states, ending up with finding chaos.

### 3.2. Intramolecular Vibrational Energy Redistribution
(IVR)

Given the statistical assumption of molecular vibrational energy distribution, the RRKM theory actually estimates the rate of unimolecular dissociation based on statistical inference at the bottle neck position on the way to products in phase space [48], using the transition state [40,41,42]. Yet, chemical dynamics has made a further step to the physical processes of intramolecular vibrational (along with rotational) energy flow (redistribution) within a molecule.

Such experimental studies were made possible by the advent of short pulse laser, which has unified two then separated subfields in molecular science, namely, spectroscopy of molecular structures and chemical reaction dynamics. As such the so-called stimulated emission pumping (SEP) spectroscopy applied to acetylene molecule H-C≡C-H (linear in shape having a triple bond) has given a huge impact [51,52,53,54]. Just briefly is outlined the essence of the experiment below. Preparing beforehand an electronically excited state of H-Ċ=Ċ-H (bent structure and longer C-C distances, double bond), they de-excited it to the high vibrational excited state on the electronic ground state by induced emission. Varying the time-width of the pumping laser, they observed very interesting vibrational emission spectra in the various hierarchical levels of resolution. Reflecting the highly vibrational excited states of polyatomic molecule, the attained spectra were extremely complicated, manifesting the chaotic feature behind. Surprisingly nevertheless, Yamanouchi succeeded to assign the spectral origins of most of the relevant peaks [52]. Among others, the most striking spectral feature is the so-called nested clump structure: Each of clump-like spectral peaks of a rather broad width is found to be composed of finer peaks attained when a longer width pulse is applied. Each of these fine peaks in turn shows another clump structure and is observed to consist of the even finer peaks with the pulse of further long width. They ascribed the nested structures to the stepwise intramolecular energy redistribution triggered by the mode change from the vibrational structure of H-Ċ=Ċ-H to that H-C≡C-H. For example, the elongated Ċ=Ċ bond releases much vibrational energy to the shorter C≡C in the first stage, which are transferred to the C-H stretching modes, which in turn coupled with the bending motion of *∠*HCC in due time ordering. More complicated energy redistribution can follow such as a very large amplitude motion of the hydrogen atoms, but we do not pursue them here. Since the spectra of the longer time-width can track the long physical process of energy redistribution. The related experiments and precise analyses have vividly shown the real-time and actual energy flow within a molecule. The notion of chaos has thus physical reality in chemical dynamics.

The energy transfer among the different vibrational modes are induced by high anharmonicity of the relevant potential functions, leading to nonlinear canonical equations, and are also due to the overlap among the Fermi resonances [55] or the Feshbach resonance [50]. (see ref. [5] for the overlap criterion of chaos.) Therefore, the relevant dynamics can readily result in chaos leading to ergodicity [56]. Many classical mechanical studies have been devoted to IVR. Yet, only recently become available extensive reviews on quantum ergodicity and intramolecular energy flow due to Leitner [57] and quantum ergodicity transition in relation to IVR in a phase space perspective by Karmakar and Keshavemurthy [58].

Incidentally, energy transfer, relaxation, and redistribution of intra- and inter-molecular systems in condensed molecular systems [55,59] and biological molecules [60,61] are also among the basic processes of many of chemical phenomena, irrespective of the extent of possible contribution by quantum chaos. Currently, molecular dynamics studies are among the main streams in these fields, but those are beyond the scope of this review.

### 3.3. Onset of Statistical Properties: Liquid-Like
Clusters

Beck and Steve Berry are the first, to the best of out knowledge, who found the critical roles of classical chaos in the statistical behavior of “melting and phase transition” appearing in structural isomerization dynamics of small atomic clusters [62]. Berry and their coworkers have achieved many fundamental works about cluster dynamics. See ref. [63] as a representative paper. We also refer to the work by Chandler et al. [64] as a pioneering study on stochastic dynamics of molecular isomerization.

Chaos not only makes the relevant phase space ergodic and mixing but also gives interesting qualitative features to the dynamics of molecules. Such properties can be typically seen in the structural isomerization dynamics of atomic clusters. Take an example from Ar${}_{7}$ cluster on the Lennard-Jones potential, which has 4 isomers as depicted in Figure 2. (Permutation symmetry counts more than 5000 isomers.) Each isomer is supported by its own potential energy basin. In a low energy, each isomer can stay within its potential basin keeping its structure as a *solid*, while at some high energy the isomerization begins to take place by surmounting the potential barriers, and further in considerably high energy frequent isomerization keeps taking place, which can be viewed as extensive large scale vibrational motions over the entire molecule or as a *liquid*-phase analog [63]. Even higher energy induces evaporation of one atom and two atomic molecule [65,66]. This entire process is hence regarded as a molecular level prototype of the first-order phase transition [62,63,67].

In the liquid-like phase, chaotic dynamics manifest themselves in the following phenomena. (1) Each classical path wanders among the potential basins for individual isomers in a random fashion (see Figure 2). The time-series with respect to the names of the isomers such as {PBP→COCT→IST→COCT→PBOP→SKEW→…} is stochastic in the Newtonian dynamics, yet a path stays for somewhat long time in each basin [68,69]. (2) Memory loss of the classical paths occurs. Despite mutually different potential barriers among the individual potential basins, they behave as though they forget which basin they came from and to which basin they are going to visit next. They seem not to care about the presence of the so-called transition state, which is the lowest energy saddle between two basins [68,69]. (3) The time scale for a trajectory to penetrate into a mixing area after getting into a potential basin is so short that trajectories readily lose the memory in the mixing area in phase space. We call this situation inter-basin mixing [70]. (4) Microcanonical temperature can be defined with which the lifetimes of the isomers are subject to an Arrhenius relation. This in turn suggests the existence of the free energy despite the nonequilibrium dynamics [71].

Behind the above chaotic dynamics we can imagine geometrical structures in phase space, referred to as the reaction tubes. Suppose that a trajectory enters a PBP basin from the neighboring COCT by passing through a divided surface (many dimensional). Then there should exist a large bundle of similar trajectories flowing from the COCT to this PBP that includes this particular trajectory. See Figure 3, which illustrates a part of the unit piece of reaction tube. Into this PBP basin schematically represented as a circle, for instance, many other reaction tubes come in from other neighboring potential basins such as those of SKEW and IST. Soon after the tubes come in, each branches frequently as schematically shown in the figure. The successive branching actually reduces the diameter exponentially as time. Indeed one can measure the exponent of the decay in the form $\exp\left(-\beta t\right)$ [70,72]. This is a geometrical background for classical paths to lose their memory from which basin they came in. Those extremely thin tubes entangle with other tubes of different histories, and they repeatedly merge with one another to thicker tubes and each eventually gets out of this basin to the neighboring ones (take the time reversal dynamics of the branching). This is how they lose the future “memory” of which basin they are going to visit next [70,72]. By construction it is obvious that the initial fat trunk of the reaction tube depicted in Figure 3 is already a mixture of finer tubes of different histories, each of those finer tubes being composed of even finer ones. Thus, the cross-section of the tube should exhibit very complicated fractal structures, along with globally periodic orbits of many different symbolic dynamics and different periods. In summary, repeated *branching and remerging* of the reaction tubes exist behind the memory loss.

The original idea of reaction tube is due to De Leon and their colleague, which they called the cylindrical manifold in the study of two dimensional two potential basin dynamics [73,74]. “Bifurcation and remerging (reconnection)” of the reaction tubes can be regarded as a basic mechanism of complex isomerization dynamics besides the baker’s transformation. There are interesting studies about the fine structure of the reaction tubes relevant to the stochastic dynamics. However we skip this aspect to quantum chaos.

## 4. Quantum Dynamics in the Quasi-Separatrix of Two Dimensional Molecular Vibration

In the early 1980s the studies of quantum chaos in terms of quantum wavepacket propagation were made rather extensively in the community of chemical physics [[75],[76],[77],[78],]. In particular, Feit and Fleck [78] are among the first, to the best of our knowledge, who numerically integrated the Schrödinger equation for wavepackets on the Hénon-Heiles Hamiltonian [27]. The scientific insights attained thus far with the accurate and the efficient methodologies developed for those studies are indispensable to date.

We herein study pure quantum chaos in terms of wavepackets and eigenfunctions confined in the separatrix on the modified Hénon-Heiles potential. This potential function is regarded as a two dimensional potential energy surface (PES) on which for molecular vibrational motion to develop.

### 4.1. Quantum Wavepackets Embedded in the Quasi-Separatrix

The Hamiltonian we adopt is
(17)$$H=\frac{1}{2m_{x}}p_{x}^{2}+\frac{1}{2m_{y}}p_{y}^{2}+\frac{1}{2}\left(x^{2}+y^{2}\right)+x^{2}\left(ay^{2}+y\right)+\frac{1}{3}y^{3}\left(by-1\right),$$
where the masses are chosen as $m_{x}=1.0087$ and $m_{y}=1.0$ and a=0.6, b=0.2with the Planck constant ℏ=0.005. The parameters have been set so as to close the dissociation channel inherent to the original Hénon-Heiles potential. The potential functions are drawn in the upper panels of Figure 4, while the Poincaré surface of section at x=0 with $p_{x}>0$ for a selected energy is drawn at the left of the lower panel. One of the regular vibrational modes lying in a torus, marked with “lib”, is drawn in the $\left(x,y\right)$-space right to the Poincaré surface of section. An appropriate energy is chosen so that the quasi-separatrix has a small to medium size so as to coexist with the set of tori of significant scales. These altogether represent *weak chaos*.

We first track a wavepacket propagating in time within the major quasi-separatrix. To do so, we extended the Poincaré surface of section to a quantum version as follows [79,80]. Given a two-dimensional wavefunction $\psi \left(x,y,t\right),$ we first take a momentum representation only in the *x*-coordinate such that
(18)$$\tilde{\psi }\left(p_{x},y,t\right)=\int_{-\infty }^{\infty }dx\exp\left(-\frac{i}{\hbar }p_{x}x\right)\psi \left(x,y,t\right).$$

Then Fourier back it using only the positive $p_{x}$ such that
(19)$$\psi^{+}\left(x,y,t\right)=\int_{0}^{\infty }dp_{x}\exp\left(\frac{i}{\hbar }p_{x}x\right)\tilde{\psi }\left(p_{x},y,t\right)$$
and put x=0 there for making a cross section at x=0. Bring it then into the Husimi representation in such a way
(20)$$\begin{array}{cc} \rho_{H}\left(q_{y},p_{y},t\right) & ={\left(2\pi \hbar \right)}^{-1}{\left(\pi \hbar \right)}^{-1/4}\\ & \times {\left|\int_{-\infty }^{\infty }dy\exp\left(-\frac{1}{2\hbar }{\left(y-p_{y}\right)}^{2}+\frac{i}{\hbar }p_{y}y\right)\psi^{+}\left(0,y,t\right)\right|}^{2} \end{array}$$

We then plot $\rho_{H}\left(q_{y},p_{y},t\right)$ in the $\left(q_{y},p_{y}\right)$-plane corresponding to the classical counterpart. This is rather a faithful extension of the classical Poincaré section.

An example of the quantum Poincaré section is exhibited in Figure 5, the upper row indicates ${\left|\psi \left(x,y,t\right)\right|}^{2}$ at selected times, while the lower row shows the corresponding quantum Poincaré section. It is clear in the quantum Poincaré section that the present packet is running on the quasi-separatrix. It is noteworthy that in upper panels of Figure 5a,c, the wavepacket stays around the unstable fixed points for a longer time, thereby leaving two significant strips in the wavefunction in the corresponding (x,y) space. These dense strips are an example of the so-called quantum scar first found by Heller [80] (see also E. J. Heller, page 547 in ref. [4]).

### 4.2. Energy Spectra and Eigenfunctions in the Quasi-Sparatrix

The next question is whether the eigenfunctions can be built in the quasi-separatrix without decaying to the stable tori. We next explore this aspect.

#### 4.2.1. Energy Screening of Quantum Wavepackets

Prior to the numerical presentation we need to show how only (or at least mostly) the separatrix eigenfunctions are extracted from the very many eigenfunctions. This is a difficult task if we resort to the straightforward method of diagonalization of the Hamiltonian matrix, which may be expanded in arbitrary basis functions. Fortunately, those wavepackets we have shown above in Figure 5 are certainly running on the quasi-separatrix, and it would be an appropriate idea to quantize those packets. Our way to do so was as follows [81].

As a standard way to obtain eigenfunctions at any chosen energy *E* from a wave packet, one may want to calculate the time-energy Fourier transform [83,84]
(21)$$\psi \left(x,E\right)={\left(2\pi \hbar \right)}^{-1}\int_{-\infty }^{+\infty }e^{+\frac{i}{\hbar }Et}\psi \left(x,t\right)\;dt\;.$$

However, this approach needs to pick extremely accurate $E^{\prime }s$ beforehand, usually by means of the Fast Fourier Transform of a correlation functions. (Incidentally see ref. [85] to extract very accurate spectrum from the FFT beyond a given grid size.) This is because the integral in Equation (Equation 21) simply vanishes in use of an inaccurate *E*. Moreover one may need to know in advance whether *E* should indeed belong to one of the quasi-separatrix eigenfunctions.

An efficient alternative to calculate eigenfunctions directly without the prior knowledge on the energy spectrum is the screening method [81]. Suppose we seek for an eigenfunction in an arbitrary narrow energy range $\left[\epsilon_{1},\epsilon_{2}\right]$. A wavepacket residing in the interval can be generated as
(22)$$\begin{array}{cc} |\Phi \left[\epsilon_{1},\epsilon_{2}\right]\rangle & =\int_{\epsilon_{1}}^{\epsilon_{2}}dE\delta \left(E-H\right)|\phi \left(0\right)\rangle \\ & =\frac{1}{2\pi }\int_{-\infty }^{\infty }dt\frac{e^{i\epsilon_{2}t}-e^{i\epsilon_{1}t}}{it}|\phi \left(t\right)\rangle , \end{array}$$
with the help of
(23)$$\delta \left(E-\hat{H}\right)={\left(2\pi \hbar \right)}^{-1}\int_{-\infty }^{+\infty }dt\;e^{\frac{i}{\hbar }\left(E-\hat{H}\right)t}\;\;,$$
where $|\phi \left(t\right)\rangle =e^{-iHt}|\phi \left(0\right)\rangle ,$ and |ϕ(0)〉 is an initial wavepacket supposed to run on the quasi-separatrix in the present purpose. If there is only one eigenstate in $\left[\epsilon_{1},\epsilon_{2}\right]$, this wavefunction $|\Phi [\epsilon_{1},\epsilon_{2}]\rangle$ should converge to to the exact one, aside from the norm. Its energy, if needed, is to be readily estimated as an expectation value. If the target energy range does not contain any eigenfunction, $|\Phi [\epsilon_{1},\epsilon_{2}]\rangle$ converges to the null. When several eigenstates are included in the range, one can step to a further elaboration, which is not difficult at all [82,86]. Incidentally, notice the faster convergence in Equation (Equation 22) than in Equation (Equation 22) due to the presence of *t* in the denominator.

#### 4.2.2. Chaotic Eigenfunctions and Spectra

Figure 6 represents an almost degenerate pair of eigenfunctions belonging to the quasi-separatrix wavefunctions. The left (right) panel represents an odd (even) function with respect to the mirror line at the x=0, with energy E=0.152094 (E=0.152500). The two energies are close to the one for the classical Poincaré section of Figure 4. As seen in the nodal structures, these are actually highly excited states. The classical motions at the two major unstable fixed pints are separated by symmetry as a left-handed (right-handed) librating motions (suggested by panel (a) and (c) in the upper row of Figure 5). However, the quantum wavepackets mix together by floating from and into those fixed point areas. Looking back at the structure of the quasi-separatrix in Figure 4, it is noticed that two wide areas around the fixed points (chaotic seas) are connected through thin canals, thus forming constrictions (see at $p_{y}=0$ in the section for instance). Therefore the wavepackets must pass through those thin channels to move from one wide chaotic sea to the other to become a global eigenfunction. In other words, those canals serve as “dynamical barriers”, which suppress the mobility of the wavepackets. The above pair of eigenfunctions in Figure 6 are generated through such dynamical barriers, and the slight difference in their energies and the odd-even symmetry separation are all the consequences of that kind of blockade or *quasi-tunneling*.

Davis and Heller found a quantum mechanical tunneling between two (or more) tori, which are separated by separatrix(es). This phenomenon is referred to as the dynamical tunneling through dynamical barrier [87,88,89]. In contrast, the pair of eigenstates observed in Figure 6 are not literally dynamical tunneling, since the canal-passing is allowed both quantum- and classical-mechanically. However, the view of the Poincaré section in Figure 5 suggests that the chaotic seas around the major fixed points reserve two major wavepackets to stay long, which are weakly separated by the thin canals thereby serving as dynamical barriers effectively. We therefore referred to this tunneling-like motion as dynamical tunneling of the second kind [80]. This phenomenon seems to be peculiar to quantum chaos, and yet we are not aware whether it has been found in actual molecular spectra.

#### 4.2.3. Dependence on the Magnitude of the Planck Constant

We are curious about the possible effects of the magnitude of the Planck constant on quantum chaos. To save the space, we refer this aspect to our original paper [79]. Yet we hereby briefly touch upon the smoothing of chaos in the larger value of *ℏ* as demonstrated in terms of the Poincaré section. The above two panels in Figure 7 come from a separatrix eigenfunction with ℏ=0.005 ($\left[E_{1},E_{2}\right]=\left[0.150.0.153\right]$), while the lower panel displays the section ℏ=0.010 ($\left[E_{1},E_{2}\right]=\left[0.150.0.156\right]$). Such quantum smoothing is not at all surprising, but the corresponding energy spectra are found not as simple (not shown here) chematic [78]. As a chemist, I suspect that the biological worlds on the globe may not be robust against a small change of the Planck constant.

### 4.3. Time-Dependent Spectrum of Very Long-Time Dynamics

The classical trajectories within the thin quasi-separatrix are generally not periodic nor quasi-periodic by definition (meaning a zero measure for the exception), and moreover they wander around circulating near one tori to another one from time to time, each torus of which represents different mode of vibration (see Figure 4). Therefore those trajectories seem to continue to undergo different vibrational motions mostly in an intermittent way [90]. Therefore it can take extremely long time to extract the associated vibrational quantum spectra from the correlation functions based on those trajectories. The resultant spectrum can be time dependent, meaning that it is dependent on which timing we begin to calculate the correlation function and on when we finish. Such an example of time-dependent spectra are seen in [79,80] This is a serious issue in vibrational spectroscopy for complicated molecules of chaotic behavior. (Recall the nested clump structure in the SEP spectrum, mentioned in Section 3.2) The structural isomerization dynamics in the liquid-like phase discussed in the previous section (Section 3.3) is actually a large amplitude vibrational motion of the extremely long time-scale associated with chaos. It is still left unresolved how to take vibrational spectra for such wandering motions from one potential basin to another, spending some long time at each basin. Proteins in molten globule state must have the similar nature to the liquid-like clusters.

## 5. Quantization of Chaos with Semiclassical Wavepackets


The mechanism of energy quantization is a landmark lying in between classical and quantum mechanics. It is therefore quite natural for semiclassical mechanics to play a major role there. Besides, the chemical dynamics quite often demands semiclassical treatment for the nuclear motion apart from chaos. There are many beautiful studies in the literature on the EBK semiclassical quantization for regular systems (see ref. [91,92] for representative examples). As for quantization of chaotic dynamics the periodic orbit theory, or the trace formula, due to Gutzwiller [35,36,37] is the central work [2,4,5,6,7,8,9,10,12]. One of the most intriguing aspects of the trace formula is its relation to the Riemann and Selberg zeta functions. However, reviewing it is by far beyond the scope of this article [93].

### 5.1. The Gutzwiller Periodic Orbit Theory

The periodic orbit theory [35,36,37] treats the semiclassical estimate of the density of states with the Feynman kernel *K* as
(24)$$D\left(E\right)=\int dq\int dtK\left(q,q,t\right)\exp\left(\frac{i}{\hbar }Et\right).$$

The theory applies the (first-order) semiclassical approximation to the Feynman kernel or its energy representation (the energy Green function) to the density of states, thereby extracting the essential roles of phase-space periodic orbits [94]. The successive applications of the stationary phase approximations give rise to the famous expression
(25)$$D\left(E\right)≃\frac{1}{{\left(2\pi \hbar \right)}^{N}}\int \int dqdp\delta \left(E-H_{cl}\left(q,p\right)\right)+\frac{1}{\pi \hbar }\sum_{\gamma }\sum_{k=1}^{\infty }\frac{T_{\gamma }\cos\left(k\left(\frac{W_{\gamma }}{\hbar }-\frac{\pi }{2}\lambda_{\gamma }\right)\right)}{\sqrt{\left|Det\left(M_{\gamma }^{k}-I\right)\right|}},$$
where the first term of the right hand side is the Thomas–Fermi (classical) density of states with the classical Hamiltonian $H_{cl}\left(q,p\right)$ arising from the periodic orbits of zero length and the second oscillatory terms represent the quantum interference among the finite periodic orbits (specified by γ) in terms of the period $T_{\gamma }$, the action integral $W_{\gamma }$, and the Monodromy matrix $M_{\gamma }^{k}$ in the transversal direction of phase-space periodic motion. These terms should be summed up with respect to the number of circulations *k* over all the possible periodic orbits. However, it is very difficult particularly in many-dimensional systems to find the necessary periodic orbits (see ref. [95] for instance) and count up their contributions accurately. Moreover, it is well known that the periodic orbits proliferate exponentially as the length of the period $T_{\gamma }$. Another intrinsic difficulty in the path summation appears in the denominator in the second term of $D\left(E\right)$. This is a direct reflection that most of the semiclassical expressions bear the serious theoretical difficulty related to divergence of the monodromy matrix, which explodes at the caustics or the turning points. The divergence is particularly prominent in chaotic systems. It is therefore often claimed that the series of periodic orbit sum in Equation (Equation 25) may not be absolutely convergent, which sometimes leaves the trace formula to be obscure in practice. Furthermore, a naive question may be cast at whether the oscillatory terms in the trace formula can indeed cancel the Thomas–Fermi density at any energy of not-to-be quantized (off-resonant energy).

It seems that there are two major opposite directions to cope with the fundamental difficulty of the trace formula. One is an attempt to recount the periodic orbit contributions in Equation (Equation 25) in such a way to attain a conditional convergence [96]. Furthermore, the convergence of the trace formula can be better considered in the complex energy plane [97]. Semiclassical mechanics of higher order *ℏ* is obviously a canonical candidate to improve the trace formula [98]. Deep mathematics are involved in these approaches.

The other one is to take an easier, simpler and more tractable approach, mainly based on the semiclassical propagation of wavepackets. Our approach below is along this line. Besides, semiclassical wavepacket theory is one of the main methodologies in chemical reaction dynamics. As such, we here present our own approach, which is intended to step further beyond semiclassical approximation to the exact one. We then attain some insights about the mechanism of energy quantization, that is, the explicit roles of not only constructive but destructive interferences in forming discrete energy levels. We also see that the energy levels in chaos can be determined without the tedious pre-exponential factor (or the Monodromy matrix) of the semiclassical kernel.

### 5.2. Wavepacket Semiclassics with Action Decomposed Function

Many semiclassical theories have been proposed in the literature [7,14,15,34,94,99,100,101,102,103,104], either starting from the Feynman kernel, WKB theory or others. Yet, we consider the following theory to clarify the dynamical hierarchical structure in the range extending from classical and semiclassical to full-quantum dynamics.

#### 5.2.1. Short Time Dynamics of the Maslov Type Wavefunction

We represent a wavefunction $\Psi \left(q,t\right)$ in the so-called Maslov type [102], such that
(26)$$\Psi \left(q,t\right)=F\left(q,t\right)\exp\left(\frac{i}{\hbar }S_{cl}\left(q,t\right)\right),$$
for a short time interval $\left[t,t+\Delta t\right]$ on a multidimensional coordinate *q* in configuration space, where $S_{cl}$ is assumed to satisfy the classical Hamilton–Jacobi (HJ) equation
(27)$$\frac{\partial S_{cl}}{\partial t}+\frac{1}{2m}{\left(\nabla S_{cl}\right)}^{2}+V=0,$$
which defines relevant classical paths during the interval, with *m* being the mass. The classical factors are thus factored out for this short interval and thereby the purely quantum factors are thus supposed to be included only in the function F(q,t)[32,33]. F(q,t) is thus referred to as Action Decomposed Function (ADF) [29,30]. Then in this interval, the Schrödinger equation for $\Psi \left(q,t\right)$ is transformed to a linear equation of motion for the complex valued amplitude function F(q,t) as
(28)$$\frac{\partial F(q,t)}{\partial t}=\left(-p\cdot \nabla -\frac{1}{2}\left(\nabla \cdot p\right)\right)F\left(q,t\right)+\frac{i\hbar }{2}\nabla^{2}F\left(q,t\right),$$
where *p* is a momentum at $\left(q,t\right),$ which is q(t), as
(29)$$p=\nabla S_{cl}\left(q,t\right).$$

We use the mass weighted coordinates so that all the masses (*m*) are scaled to unity m=1, and thereby *p* is set to be numerically equivalent to the corresponding velocity *v*. In transforming the Euler picture to the Lagrange by defining
(30)$$\frac{D}{Dt}=\frac{\partial }{\partial t}+v\cdot \nabla$$

Equation (Equation 28) reads
(31)$$\frac{D}{Dt}F\left(q,t\right)=\left[-\frac{1}{2}\left(\nabla \cdot p\right)+\frac{i\hbar }{2}\nabla^{2}\right]F\left(q,t\right).$$

$F(q-q_{cl}\left(t\right),t)$ is hence carried along a classical path $q_{cl}\left(t\right)$ in this interval. Then the Trotter decomposition [94] valid for a very short time–interval Δt gives
(32)$$\begin{array}{c} F\left(q-q_{cl}\left(t+\Delta t\right),t+\Delta t\right)\\ ≃\exp\left[\frac{i\hbar }{2}\Delta t\nabla^{2}\right]\exp\left[-\frac{1}{2}\left(\nabla \cdot p\right)\Delta t\right]F\left(q-q_{cl}\left(t\right),t\right)\\ ≃\exp\left[-\frac{1}{2}\left(\nabla \cdot p\right)\Delta t\right]\exp\left[\frac{i\hbar }{2}\Delta t\nabla^{2}\right]F\left(q-q_{cl}\left(t\right),t\right) \end{array}$$
and represents the quantum effects in a product between the momentum gradient ($\exp\left[-\frac{1}{2}\left(\nabla \cdot p\right)\Delta t\right]$) and the pure quantum diffusion ($\exp\left[\frac{i\hbar }{2}\Delta t\nabla^{2}\right]$). The theory is exact except for the Trotter decomposition, provided that the classical action $S_{cl}\left(q,t\right)$ exists.

#### 5.2.2. Semiclassical Approximation

Equation (Equation 32) is first reduced to a semiclassical level of approximation by considering the momentum gradient part only, which results in
(33)$$\begin{array}{c} F\left(q-q_{cl}\left(t+\Delta t\right),t+\Delta t\right)\\ ≃\exp\left[-\frac{1}{2}\left(\nabla \cdot p\right)\Delta t\right]F\left(q-q_{cl}\left(t\right),t\right) \end{array}$$

Notice that the Planck constant does not yet appear in this level. The time evolution of it is approximately given by estimating the momentum gradient term as follows [29]
(34)$$\begin{array}{c} {\left.F(q-q(t+\Delta t),t+\Delta t)\right|}_{q=q_{cl}\left(t+\Delta t\right)}\\ ={\left.\exp\left[-\frac{1}{2}\int_{t}^{t+\Delta t}dt\sum_{k}\frac{\partial p_{k}}{\partial q_{k}}\right]F\left(q-q_{cl}\left(t\right),t\right)\right|}_{q=q_{cl}\left(t\right)}\\ ={\left.{\left(\frac{\sigma (t)}{\sigma (t+\Delta t)}\right)}^{1/2}F\left(q-q_{cl}\left(t\right),t\right)\right|}_{q=q_{cl}\left(t\right)} \end{array}$$
or
(35)$${\left.\sigma {\left(t+\Delta t\right)}^{1/2}F\left(q-q_{cl}\left(t+\Delta t\right),t+\Delta t\right)\right|}_{q=q_{cl}\left(t+\Delta t\right)}={\left.\sigma {\left(t\right)}^{1/2}F\left(q-q_{cl}\left(t\right),t\right)\right|}_{q=q_{cl}\left(t\right)},$$
where the deviation determinant $\sigma \left(t\right)$ is expressed as
(36)$$\sigma \left(t\right)=\prod_{i=1}^{N}\wedge \left(q_{cl}^{i}\left(t\right)-q_{cl}\left(t\right)\right),$$
which is an *N*-dimensional orientable tiny volume surrounding the point $q_{cl}\left(t\right)$ in configuration space, with $q_{cl}^{i}\left(t\right)$ (i=1,…,N) being the configuration space point of an *i*th classical trajectories nearby the reference path $q_{cl}\left(t\right)$. See Figure 8 for geometrical definition of $\sigma \left(t\right)$. Since $\sigma \left(t\right)$ arises from the orientable volume, the square root of it gives rise to the part of the so-called Maslov phase [105]. (see ref. [106] for another geometrical meaning of the Maslov phase.)

The above results can be represented in a symmetric manner by defining “square root” of the volume element [107]. Suppose that an arbitrary function $f\left(q_{t}\right)$, where $q_{t}$ is the coordinate system in configuration space, is mapped from the original point $q_{0}$ at time 0 by the classical flow $q_{0}\to$$q_{t}$. An integral also mapped is represented as
(37)$$I=\int f\left(q{}_{t},t\right)dq_{t}=\int f\left(q{}_{t},t\right)dq_{t}^{1/2}dq_{t}^{1/2*}$$
with
(38)$$dq_{t}^{1/2}\equiv \exp\left[\frac{i\pi M\left(q_{0},q_{t}\right)}{2}\right]{\left|dq_{t}\right|}^{1/2},$$
where $dq_{t}^{1/2*}$ is intended to be the complex conjugate to $dq_{t}^{1/2}$, and $M\left(q_{0},q_{t}\right)$ is the Maslov index counting the number of the change of sign of $\sigma \left(t\right)$ of Equation (Equation 36). The semiclassical conservation rule is then written up to the phase as [107]
(39)$$F\left(q{}_{t},t\right)dq_{t}^{1/2}=F\left(q{}_{0},0\right)dq_{0}^{1/2}.$$

This is rather similar to the law of the Poincaré-Cartan absolute invariance in phase space [108].

The momentum gradient $\sum_{k}\partial p_{k}/\partial q_{k}$ can diverge, when it happens to hold $\sigma \left(t\right)=0$ in Equation (Equation 35), where a classical path come to a focal point either in configuration or momentum space. This is the origin of the difficulty of the theories of semiclassical level. Yet, such a singularity can be removed by the quantum diffusion below [29].

#### 5.2.3. Gaussian Wavepacket Approximation

We next proceed to take account of the quantum diffusion term. In doing so, we here survey the dynamics of a Gaussian wavepacket among other possible choices of F(q,t). Heller introduced the dynamics of a Gaussian function as early as in 1975 by assuming that the potential function at the Gaussian center is truncated up to the quadratic terms [109].

We insert a Gaussian of the following form into the general scheme of ADF Equation [29],
(40)$$G\left(q-q_{cl}\left(t\right),t\right)=g\left(t\right)\exp\left[-\frac{1}{c\left(t\right)+id\left(t\right)}{\left(q-q_{cl}\left(t\right)\right)}^{2}\right],$$
with both $c\left(t\right)$ and $d\left(t\right)$ being real-valued. Then some (rigorous) manipulations give
(41)$$c\left(t+\Delta t\right)=c\left(t\right){\left(\frac{\sigma \left(t+\Delta t\right)}{\sigma \left(t\right)}\right)}^{2}$$
and
(42)$$d\left(t+\Delta t\right)=d\left(t\right){\left(\frac{\sigma \left(t+\Delta t\right)}{\sigma \left(t\right)}\right)}^{2}+\frac{2\hbar \Delta t}{m}.$$
or shows they satisfy the following differential equation, respectively,
(43)$$\dot{c}\left(t\right)=2\frac{\dot{\sigma }\left(t\right)}{\sigma \left(t\right)}c\left(t\right).$$
and
(44)$$\dot{d}\left(t\right)=2\frac{\dot{\sigma }\left(t\right)}{\sigma \left(t\right)}d\left(t\right)+\frac{2\hbar }{m}.$$

Note that the Planck constant appears only in this level of dynamical hierarchy. This normalized Gaussian form are propagated in time quite accurately, under the dynamical circumstances where the Gaussian may remain Gaussian. Yet, the theory needs additional procedures when the dynamics under study faces difficult situations like wavepacket bifurcation [110].

Further, one can find an interesting internal structure in the Gaussian packet. Define ζ(t) then such that
(45)$$d\left(t\right)=\zeta \left(t\right)\sigma \left(t\right),$$
and insert this back into Equation (Equation 44) to obtain
(46)$$\sigma \left(t\right)\dot{\zeta }\left(t\right)-\dot{\sigma }\left(t\right)\zeta \left(t\right)=\frac{2\hbar }{m},$$
which is a Wronskian relation between $\sigma \left(t\right)$ and ζ(t). This suggests that *something* is rotating to maintain the packet in the quantum realm [29]. However, we step aside from this subject to be back to the energy quantization problem [110].

### 5.3. On the Quantization Condition: Creation and Elimination of the
Spectral Peaks by the Phases

It is quite often that only the to-be-quantized (resonant) energy peaks are under focus without paying much attention to to-be-nullified part of $D\left(E\right)$ at off-resonant energy areas. We therefore reconsider here the basic mechanisms of evolution of the resonant peaks and those of suppression of the off-resonant counterparts.

#### 5.3.1. Prior Quantization Conditions Based on the Periodic Orbits

To reconsider the mechanism of quantization from the view point of wavepacket dynamics, we get back to a little more elementary stage of the theory. Consider a spectral density function
(47)$$P\left(E\right)=\int_{PO}\frac{dq_{0}dp_{0}}{\hbar^{N}}P_{tr}\left(E_{cl}\left(q_{0},p_{0}\right);E\right),$$
for an *N*-dimensional system, where the pre-spectral density is defined as
(48)$$\begin{array}{c} P_{tr}\left(E_{cl}\left(q_{0},p_{0}\right);E\right)\\ =\lim_{L\to \infty }\frac{1}{L}\int_{0}^{L}dtR\left(q_{t},q_{0},t\right)\exp\left[\frac{i}{\hbar }S\left(q_{t},q_{0},t\right)-i\frac{\pi }{2}M\left(q_{0},q_{t}\right)\right]\exp\left(\frac{i}{\hbar }Et\right) \end{array}$$
which is a correlation function starting from a classical point $\left(q_{0},p_{0}\right)$ on a manifold of energy $E_{cl}$. $M(q_{0},q_{t})$ is the Maslov index. $R(q_{t},q_{0},t)$ represents an overlap integral between a wavepacket at t=0 and that of later time *t*. We implicitly assume that the wavepacket adopted here has been taken from the above ADF framework, Equation (Equation 26), yet we do not specify the functional form of it in this stage and it does not have to be a Gaussian. The Fourier transform in $P_{tr}$ eventually gives rise to the energy spectrum in Equation (Equation 47). In the following analysis, we assume that (1) only the periodic orbits are to be considered in $P_{tr}\left(E_{cl}\left(q_{0},p_{0}\right);E\right)$ as the symbol $\int_{PO}$ indicates, which is a natural consequence of the stationary-phase condition, (2) the phases in Equation (Equation 47) are just made up with the standard action integral and the associated Maslov phases alone, and (3) the amplitude factor $R(q_{t},q_{0},t)$ is always finite and smooth along the trajectories [111,112].

#### 5.3.2. Peaks Arising from Individual Orbits and an Extended
Quantization Condition

In the time-integration of $P\left(E\right)$ in Equation (Equation 48), one can apply the stationary phase approximation (SPA) to obtain
(49)$$\frac{\partial }{\partial t}\left[S\left(q,q_{0};t\right)+Et\right]=-E_{cl}+E=0,$$
which indicates that classical trajectories with $E_{cl}$ should make a major contribution to $P_{tr}\left(E_{cl}\left(q_{0},p_{0}\right);E\right)$, which is already well-known [35,36,37,113,114]. And $P_{tr}\left(E_{cl}\left(q_{0},p_{0}\right);E\right)$ can be viewed as a function of both *E* and $E_{cl}\left(q_{0},p_{0}\right)$.

We first survey the oscillatory property of the integrand of $P_{tr}\left(E_{cl}\left(q_{0},p_{0}\right);E\right)$. With the classical action integral
(50)$$S\left(q_{t},q_{0},t\right)=\int_{q_{0}}^{q_{t}}p\left(q;E_{cl}\left(q_{0},p_{0}\right)\right)dq-E_{cl}\left(q_{0},p_{0}\right)t,$$
we define
(51)$$\Lambda \left(t;q_{0},p_{0}\right)=R\left(q_{t},q_{0},t\right)\exp\left[\frac{i}{\hbar }{\left.\int_{q_{0}}^{q_{t}}p\left(q;E_{cl}\left(q_{0},p_{0}\right)\right)dq\right|}_{\mathrm{PO}}-i\frac{\pi }{2}M\left(q_{0},q_{t}\right)\right],$$
to rewrite $P_{tr}\left(E_{cl}\left(q_{0},p_{0}\right);E\right)$ in a simplified fashion as
(52)$$P_{tr}\left(E_{cl}\left(q_{0},p_{0}\right);E\right)=\int dt\Lambda \left(t;q_{0},p_{0}\right)\exp\left[\frac{i}{\hbar }\left(E-E_{cl}\left(q_{0},p_{0}\right)\right)t\right].$$

Note that both $\int_{q_{0}}^{q_{t}}p\left(q;E_{cl}\left(q_{0},p_{0}\right)\right)dq$ and $\frac{\pi }{2}M\left(q_{0},q_{t}\right)$ have the following time dependence on the periodic orbits
(53)$$\left(\mathrm{constant}\right)\times t+\left(\mathrm{oscillatory}\;\mathrm{part}\right).$$

Here the constant part is calculated out as an average of $\int_{q_{0}}^{q_{t}}p\left(q;E_{cl}\left(q_{0},p_{0}\right)\right)dq$ and $\frac{\pi }{2}M\left(q_{0},q_{t}\right)$ in $\Lambda \left(t;q_{0},p_{0}\right)$ of Equation (Equation 51) for the interval of, say, $\left[q_{0},q_{T}\right]$. Let the period of the periodic orbit under consideration be $T\left(q_{0},p_{0}\right)$. Then we define a quantity $X\left(q_{0},p_{0}\right)$ such that
(54)$$\frac{1}{T\left(q_{0},p_{0}\right)}\left[\frac{1}{\hbar }\oint_{q_{0}}^{q_{T}}p\left(q;E_{cl}\left(q_{0},p_{0}\right)\right)dq-\frac{\pi }{2}M\left(q_{0},q_{T}\right)\right]=X\left(q_{0},p_{0}\right),$$
and extract $X\left(q_{0},p_{0}\right)\times t$ from the phase part of (Equation 51) (that is Equation (Equation 53)) in such a way that
(55)$$\frac{1}{\hbar }\int_{q_{0}}^{q_{t}}p\left(q;E_{cl}\left(q_{0},p_{0}\right)\right)dq-\frac{\pi }{2}M\left(q_{0},q_{t}\right)-X\left(q_{0},p_{0}\right)t.$$

Then this function should appears to be an exactly periodic function. Furthermore, since the preexponential factor $R(q_{t},q_{0},t)$ is also assumed to be periodic of $T\left(q_{0},p_{0}\right)$,
(56)$$\Lambda \left(t;q_{0},p_{0}\right)\exp\left[-iX\left(q_{0},p_{0}\right)t\right]$$
is periodic with the period $T\left(q_{0},p_{0}\right)$ as a whole. Then one can expand the function $\Lambda \left(t;q_{0},p_{0}\right)$ in a Fourier series such that
(57)$$\Lambda \left(t;q_{0},p_{0}\right)=\sum_{n=-\infty }^{\infty }c_{n}\exp\left(-i\frac{2n\pi }{T\left(q_{0},p_{0}\right)}t+iX\left(q_{0},p_{0}\right)t\right),$$
where
(58)$$c_{n}=\frac{1}{T\left(q_{0},p_{0}\right)}\int_{0}^{T}dt\Lambda \left(t;q_{0},p_{0}\right)\exp\left[i\frac{2n\pi }{T\left(q_{0},p_{0}\right)}t-iX\left(q_{0},p_{0}\right)t\right].$$

Here we take a limit in the time integral for $P_{tr}\left(E_{cl}\left(q_{0},p_{0}\right);E\right)$ in Equation (Equation 48) up to sufficiently long time *L* and rewrite it with the help of Equation (Equation 57) as
(59)$$\begin{array}{c} P_{tr}\left(E_{cl}\left(q_{0},p_{0}\right);E\right)\\ \equiv \frac{1}{L}\int_{0}^{L}dt\Lambda \left(t;q_{0},p_{0}\right)\exp\left[\frac{i}{\hbar }\left(E-E_{cl}\left(q_{0},p_{0}\right)\right)t\right]\\ =\frac{1}{L}\sum_{-\infty }^{\infty }c_{n}\frac{-i\hbar T\left(q_{0},p_{0}\right)}{-2n\pi \hbar +\left(E-E_{cl}\left(q_{0},p_{0}\right)+X\hbar \right)T\left(q_{0},p_{0}\right)}\\ \times \left(\exp\left[\frac{iL}{\hbar T\left(q_{0},p_{0}\right)}\left(-2n\pi \hbar +\left(E-E_{cl}\left(q_{0},p_{0}\right)+X\hbar \right)T\left(q_{0},p_{0}\right)\right)\right]-1\right) \end{array}$$

The correlation function $R(q_{t},q_{0},t)$ is assumed to be so smoothly varying (or much less oscillatory than the phase factor) that it can be practically factored out from the Fourier expansion with respect to the phases.

It is now clear from Equation (Equation 59) that only the denominator in this expression satisfying the condition
(60)$$\left(E-E_{cl}\left(q_{0},p_{0}\right)+X\hbar \right)T\left(q_{0},p_{0}\right)=2n\pi \hbar$$
will lead to a peak in $P_{tr}\left(E_{cl}\left(q_{0},p_{0}\right);E\right)$ of the height of $c_{n}$. The substitution of $X\left(q_{0},p_{0}\right)$ with that of Equation (Equation 54) gives rise to a condition
(61)$$\left(E-E_{cl}\left(q_{0},p_{0}\right)\right)T\left(q_{0},p_{0}\right)+\left[{\left.\int_{q_{0}}^{q_{t}}p\left(q;E_{cl}\left(q_{0},p_{0}\right)\right)dq\right|}_{\mathrm{PO}}-\frac{\pi }{2}M\left(q_{0},q_{T}\right)\hbar \right]=2n\pi \hbar .$$

This is a quantization (resonance) condition for $P_{tr}\left(E_{cl}\left(q_{0},p_{0}\right);E\right)$ (but not for $P\left(E\right)$), which we call the *prior* quantization condition [111,115]. Not only *E* but $\left(q_{0},p_{0}\right)$ has to fulfill this condition to make a peak in $P_{tr}\left(E_{cl}\left(q_{0},p_{0}\right);E\right)$. Although $P_{tr}\left(E_{cl}\left(q_{0},p_{0}\right);E\right)$ has far more peaks than $P\left(E\right)$ itself, many of these peaks are to be eliminated by *destructive interference* upon integration over the trajectory space, which will be discussed below.

Yet, the true peaks should be contained in the set of peaks fulfilling Equation (Equation 61). Note also that the additional condition of Equation (Equation 49) nullifies
(62)$$\left(E-E_{cl}\left(q_{0},p_{0}\right)\right)T\left(q_{0},p_{0}\right)$$
in Equation (Equation 61), thereby reducing it directly to the EBK-like condition.

#### 5.3.3. Destructive Interference Suppressing the Spurious Energy Peaks:
Another Essential Role of the Phases for Quantization

The off-resonant trajectories should suppress the values of $P\left(E\right)$ to the order of O$\left(\hbar \right)$ due to the Riemann-Lebesgue lemma. However, there can exist many quasi-resonant trajectories, which would form spurious peaks in $P_{tr}\left(E_{cl}\left(q_{0},p_{0}\right);E\right)$. These peaks are anticipated to be cancelled away by destructive interference with many other anonymous trajectories. As such there are two basic case in which to suppress virtual peaks as follows. (We refer to the original papers, ref. [111,112], for more details along with numerical presentations.)

##### Case in which E Is Slightly Shifted from a True Eigenvalue
(Near-Resonance Peaks)

Suppose there is a peak satisfying
(63)$$\left[\int_{q_{0}}^{q_{t}}p\left(q;E_{cl}\left(q_{0},p_{0}\right)\right)dq-\frac{\pi }{2}M\left(q_{0},q_{T}\right)\hbar \right]=2n\pi \hbar$$
at an energy $E_{cl}\left(q_{0},p_{0}\right)=E_{x}$. It is expected theoretically that the contribution from nearby orbits having $E_{cl}^{\prime }\left(q_{0}^{\prime },p_{0}^{\prime }\right)$ should make $P_{tr}\left(E_{cl}^{\prime }\left(q_{0}^{\prime },p_{0}^{\prime }\right);E_{x}\right)$ very small to the order of O$\left(\hbar \right)$. However, if $E_{cl}^{\prime }\left(q_{0}^{\prime },p_{0}^{\prime }\right)$ happens to be sufficiently close to the $E_{x}=E_{cl}\left(q_{0},p_{0}\right)$, the peak is split into “*a pair of* very *high spurious* peaks” that contain the peak of $P_{tr}\left(E_{cl}^{\prime }\left(q_{0}^{\prime },p_{0}^{\prime }\right);E_{x}\right)$ in between. These peaks constitute a continuous width near the true peak. (For numerical and graphical examples, see [111].)

##### Case of Generation of Harmonics in E

There is yet another mechanism to generate spurious peaks. A trajectory with $E_{cl}\left(q_{0},p_{0}\right)$ happens to satisfy the resonance condition of Equation (Equation 63) with some “quantum number” *n* and the energy $E_{x}$. Then, it is possible that the prior quantization condition of Equation (Equation 61) may be satisfied by another set of $\left(n^{\prime },E_{x}^{\prime }\right)$. They must be simply the harmonics of the correct peak of $\left(n,E_{x}\right)$ [111]. These harmonics also should be eliminated by off-resonant trajectories in terms of the appropriate destructive interference.

Having presented the specific roles of off-resonant trajectories to suppress the spurious peaks, we are concerned whether the trace formula implements such a function in it.

### 5.4. Amplitude-Free Energy Quantization of Classical Chaos

One of the theoretical and numerical difficulties of the trace formula manifests itself explicitly in the denominator of Equation (Equation 25) of the second term, which comes from the amplitude factor of the energy Greens function. In the above analysis of the energy spectrum, from Equations (Equation 48)–(Equation 61), no significant role was found for the amplitude factor $R(q_{t},q_{0},t)$ to play. We hence try to pursue a possible energy quantization disregarding the absolute value of the amplitude factor, but taking the Maslov phase into account.

To explicitly monitor the sensitivity of the amplitude factor to the variation of the preexponential factor, we first make the following preliminary calculations [116], in which the amplitude factor is rescaled intentionally as follows.
(64)$${}^{}\begin{array}{cc} C_{S}^{ADF}\left(t\right) & =\int dq_{0}F^{*}\left(q_{t},0\right)F\left(q_{0},0\right){\left|\frac{\partial q_{t}}{\partial q_{0}}\right|}^{1-s}\\ & \times \exp\left[\frac{i}{\hbar }S_{1}\left(q_{t},q_{0},t\right)+\frac{i}{\hbar }p_{0}\cdot \left(q_{0}-q_{t}\right)-\frac{i\pi M}{2}\right] \end{array}$$
where $C_{s}^{ADF}\left(t\right)$ is a correlation function using a primitive Gaussian function, and an artificial scaling parameter *s* has been introduced. The value s=0.5 reduces $C_{s}^{ADF}$ to the original semiclassical correlation function, while there is no physical justification to choose other values. Yet, it numerically turns out that values up to at least s=1.5 give correct energy positions in the resultant spectra [116]. In particular the choice of s=1.0 implies that one can abandon the semiclassical amplitude factor as ${\left|\frac{\partial q_{t}}{\partial q_{0}}\right|}^{0}$, aside from the Maslov phase, to obtain the spectral positions.

After somewhat extensive analysis and numerical studies, we have confirmed that the spectrum positions are accurately reproduced through an amplitude correlation function [117,118], of the form
(65)$$C_{p_{0}}^{III}\equiv \int dq_{0}F^{*}\left(q_{t}\right)F\left(q_{0}\right)\exp\left(-\frac{i}{\hbar }p_{0}q_{t}\right)\exp\left[\frac{i}{\hbar }S_{2}\left(q_{t},p_{0},t\right)-\frac{i\pi M}{2}\right],$$
where $p_{0}$ is an initial momentum to be scanned to cover the entire spectrum, and $F(q_{t})$ is a (frozen) Gaussian function to be carried from the initial position $q_{0}$ [115,116]. We only need the way of calculation of the Maslov index as exemplified in Figure 8. It turns out that the amplitude-free correlation function $C_{p_{0}}^{III}$ works quite well even in chaotic regime, provided that the semiclassical dynamics under study is essentially dominated by the phases. An example of chaotic spectrum in chaotic region for the Hamiltonian
(66)$$H=\frac{1}{2m_{x}}p_{x}^{2}+\frac{1}{2m_{y}}p_{y}^{2}+\frac{1}{2}\left(x^{2}+y^{2}\right)+x^{2}\left(0.6y^{2}+y\right)+\frac{1}{3}y^{3}\left(0.2y-1\right)+0.1x$$
with $m_{x}=1.0087$ and $m_{y}=1.0$ is shown in Figure 9, where we have adopted ℏ=0.005 and $E_{cl}≃0.157.$ (Note that for a large *ℏ*, the quality of semiclassical quantization is accordingly deteriorated and should not be used.)

The partial success of the amplitude-free quantization without paying full attention to the semiclassical amplitude factors, except for the Maslov phase, suggests that one does not necessarily have to faithfully track the every fine complexity of the Lagrange manifold behind classical chaos in order to perform semiclassical mechanics, partly due to the quantum smoothing.

### 5.5. Chaos Mediated by Dynamical Tunneling

Tunneling effect is one of the key phenomena to characterize chemical reactions not only in very low temperature circumstances but even in excited states. It is now widely accepted that complex semiclassical theory is very effective in the study of quantum tunneling [119]. However, classical chaos generally makes the structures of the Lagrange manifold and associated caustics very complicated in multidimensional complex space. Besides, the choice of the stationary paths there is not necessarily achieved as in the Feynman kernel over the real paths. Shudo and Ikeda have developed a pioneering work to lay out and cope with the intricate yet beautiful tunneling paths in the complex domain [120,121]. We here present a simple-minded idea on semiclassical tunneling by introducing the parity of motion, which splits phase space into many mutually exclusive nonclassical sub-spaces [122,123].

#### 5.5.1. Semiclassical Tunneling Theory

We first need to formulate a tunneling theory in a semiclassical context [122,123]. Let us begin with Feynman’s path integral representation of the kernel
(67)$$K\left(x,\Delta t;y\right)={\left(\frac{m}{2\pi i\hbar \Delta t}\right)}^{N/2}\exp\left(\frac{i\Delta t}{\hbar }\left[\frac{m}{2}{\left(\frac{x-y}{\Delta t}\right)}^{2}-V\left(\frac{x+y}{2}\right)\right]\right)$$
or in a more limited alternative
(68)$$K\left(q+\Delta q,\Delta t;q\right)={\left(\frac{m}{2\pi i\hbar \Delta t}\right)}^{N/2}\exp\left(\frac{i\Delta t}{\hbar }\left[\sum_{k}\frac{m_{k}}{2}{\left(\frac{\Delta q_{k}}{\Delta t}\right)}^{2}-V\left(q\right)\right]\right),$$
where Δq is not necessarily as short as Δt, and V(q) is an arbitrary potential function. In the path integration, the Euler variational principle
(69)$$\frac{d}{dt}\left(\frac{\partial L}{\partial {\dot{q}}_{k}}\right)-\frac{\partial L}{\partial q_{k}}=0,$$
with the Lagrangian
(70)$$\begin{array}{cc} L & =\sum_{k}\frac{m_{k}}{2}{\left(\frac{\Delta q_{k}}{\Delta t}\right)}^{2}-V\left(q\right)\\ & =\sum_{k}\frac{m_{k}}{2}{\left({\dot{q}}_{k}\right)}^{2}-V\left(q\right)\;\;\mathrm{as}\;\Delta q\to 0\;\mathrm{with}\;\Delta t\to 0 \end{array}$$
brings about the stationarity in the integral [94]. It is seen though that the kernel of Equation (Equation 68) is invariant to a set of parameters (constant) $\sigma_{k}$ that are introduced in a manner that
(71)$$K\left(q+\Delta q,\Delta t;q\right)={\left(\frac{m}{2\pi i\hbar \Delta t}\right)}^{N/2}\exp\left(\frac{i\Delta t}{\hbar }\left[\sum_{k}\frac{m_{k}}{2}{\left(\sigma_{k}\frac{\Delta q_{k}}{\Delta t}\right)}^{2}-V\left(q\right)\right]\right)$$
as long as
(72)$$\sigma_{k}^{2}=1.$$

Define accordingly the corresponding Lagrangian $L\left(\sigma \right)$ as
(73)$$L\left(\sigma \right)=\sum_{k}\frac{m_{k}}{2}{\left(\sigma_{k}{\dot{q}}_{k}\right)}^{2}-V\left(q\right)$$
which should satisfy the stationary condition
(74)$$\frac{d}{dt}\left(\frac{\partial L(\sigma )}{\partial {\dot{q}}_{k}}\right)-\frac{\partial L(\sigma )}{\partial q_{k}}=0.$$

An additional set of the Newtonian equations then results
(75)$$m_{k}{\ddot{q}}_{k}=-\sigma_{k}\frac{\partial V(q)}{\partial q_{k}}.$$

Nothing happens when all $\sigma_{k}=1$. If all the $\sigma_{k}$ is −1, the paths emerged from Equation (Equation 75) are equivalent to the so-called instanton path [94]. We refer to $\sigma_{k}$ as the parity of motion. A variety of the set $\left\{\sigma_{k}\right\}$ gives the various combinations of non-tunneling and tunneling paths in the real time. Thus, the sets of $\left\{\sigma_{k}\right\}$ defines the individual phase spaces, which we call sheets. No communication exists among them as long as the parity sets are frozen.

We can move to the Hamiltonian framework by defining
(76)$$p_{k}=\sqrt{\sigma_{k}}\dot{q_{k}},$$
where $p_{k}$ is pure imaginary for the negative parity. The corresponding canonical equations of motion
(77)$$\sigma_{k}\dot{\bar{p_{k}}}=-\frac{\partial H(\sigma )}{\partial q_{k}}\;\;\;\;\mathrm{and}\;\;\;\;\sigma_{k}\dot{q_{k}}=\frac{\partial H(\sigma )}{\partial \bar{p_{k}}},$$
is defined with the modified Hamiltonian
(78)$$H\left(\sigma \right)=\sum_{k}\frac{\sigma_{k}}{2m_{k}}{\bar{p_{k}}}^{2}+V\left(q\right).$$

Setting $\sigma_{k}=-1$ is effectively equivalent to flipping the mass $m_{k}$ from positive to negative value.

To see the invariance property inherent to the Hamilton system, the following coordinates are convenient, that is
(79)$$Q_{k}=\sqrt{\sigma_{k}}q_{k}\;\;\;\;\;\;\;\mathrm{and}\;\;\;\;\;\;P_{k}=\sqrt{\sigma_{k}}{\bar{p}}_{k},$$
in which the new coordinates $Q_{k}$’s in configuration space are also pure imaginary when their parities are negative. Then Equation (Equation 77) are transformed to
(80)$${\dot{P}}_{k}=-\frac{\partial H(\sigma )}{\partial Q_{k}}\;\;\;\;\;\;\;\mathrm{and}\;\;\;\;\;\;{\dot{Q}}_{k}=\frac{\partial H(\sigma )}{\partial P_{k}}.$$

Due to the Poincaré-Cartan theorem of absolute invariance [108], the symplectic structure the 2-form
(81)$$\omega^{2}=\sum_{k}dP_{k}\wedge dQ_{k}$$
is conserved and the Stokes theorem allows to define the path-action $S_{cl}$, that is,
(82)$$S_{cl}=\int \sum_{k}P_{k}dQ_{k}-H\left(Q,P\right)dt.$$

In order to adopt these paths in semiclassical mechanics, the *Q*-coordinates should be rotated back to *q*-coordinates keeping *P* in such a way that
(83)$$\left(Q,\;P\right)\to \left(q,\;P\right).$$

Then we have an action integral
(84)$$\begin{array}{cc} S_{cl}\left(Q,P\right) & \to S_{\mathrm{trl}}\left(q,P\right)=\int \sum_{k}Pdq_{k}-H\left(\sigma \right)dt\\ & =\sum_{k}\int \sqrt{\sigma_{k}}{\bar{p}}_{k}dq_{k}-H\left(\sigma \right)dt \end{array}$$
and a semiclassical wavefunction for tunneling is formally written as
(85)$$\psi =A\exp\left(\frac{i}{\hbar }S_{\mathrm{tnl}}\right),$$
where the pre-exponential factor *A* represents an amplitude. Since $S_{\mathrm{tnl}}$ in Equation (Equation 84), is generally complex-valued, we rewrite $S_{\mathrm{tnl}}$ in the standard form as
(86)$$S_{\mathrm{tnl}}=S_{R}+iS_{I}.$$

Insertion of Equation (Equation 86) into Equation (Equation 85) gives
(87)$$\psi =A\exp\left(-\frac{1}{\hbar }\left|S_{I}\right|\right)\exp\left(\frac{i}{\hbar }S_{R}\right),$$
where the phase convention to $S_{\mathrm{tnl}}$ is taken so that $S_{I}$ becomes positive. Thus, the damping of the norm due to tunneling is estimated as
(88)exp−2ImStnlℏ,
which gives a natural definition of the relevant tunneling probability.

The tunneling paths induce additional phases when they are incorporated into the semiclassical Feynman kernel such that [124]
(89)$$K\left(q_{t},t;q_{s},s:\sigma \right)=\sum_{\alpha }{\left(\frac{1}{-2\pi i\hbar }\right)}^{\frac{N}{2}}\frac{1}{\sqrt{\left|\frac{\partial q_{t}}{\partial {\bar{p}}_{s}}\right|}}\exp\left(\frac{i}{\hbar }S_{\mathrm{tnl}}+i\phi \right),$$
where *N* is the number of dimensions and the additional phase ϕ is given by
(90)$$\phi =-\frac{\pi }{2}N_{f}+\frac{\pi }{4}N_{p}$$
at an entrance of a tunneling path and
(91)$$\phi =-\frac{\pi }{2}N_{f}-\frac{\pi }{4}N_{p}$$
at an exit of a tunneling path. Here $N_{p}$ is the number of the negative parities while $N_{f}$ is the sum of the multiplicities of the zeros of $\partial q_{f}/\partial {\bar{p}}_{i}=0$ along the trajectory, which gives the Maslov index.

##### Connection Problem


Tunneling paths can arise only upon transferring from $\left\{\sigma_{k}=1\right\}$ to another, typically only one of $\sigma_{k}$ is negative. This is because as the number of negative parities increases the damping probability in Equation (Equation 88) also become large. Those paths on the different sheets should be connected smoothly. To ensure smoothness, we change the sign of one of the parities at a caustic in the direction normal to the caustic line. The caustics, at which the density of the paths becomes very high (actually infinite in the primitive semiclassical approximation), are defined with the so-called Jacobi field [94] in such a way that
(92)$$det|\frac{\partial \vec{q_{f}}}{\partial \vec{p_{i}}}|=0,$$
where $\vec{p_{i}}$ is a real-valued (quasi-)momentum vector for a given parity set {$\sigma_{k}=\pm 1$} at the initial point on each sheet and $\vec{q_{f}}$ is the final points of a trajectory on the same sheet. The tunneling path can get back to the Newtonian sheet of $\left\{\sigma_{k}=1\right\}$ in a similar manner [125].

The stationary phase condition in the semiclassical approximation to the Feynman kernel requires that the quantum phase $S_{HJ}$ in Equation (Equation 89) should be connected smoothly at the connection point. This requirement is expressed as
(93)$$\frac{\partial S_{HJ}}{\partial q}\;\;\left(\mathrm{in}\;\mathrm{classical}\;\mathrm{region}\right)=\frac{\partial S_{\mathrm{tnl}}}{\partial q}\;\;\left(\mathrm{in}\;\mathrm{tunnel}\;\mathrm{region}\right),$$
which simply implies
(94)$$\bar{p_{k}}=-\bar{p_{k}}$$
at the connection point. Obviously,
(95)$$\bar{p_{k}}=0$$
is the unique solution [126]. Therefore, to connect the wavefunctions smoothly at the turning points in the relevant direction *k*, we may change the parity there (or parities depending on the degeneracy of turning points) to the direction of zero momentum.

Tunneling phenomena are quite important in many chemical reaction dynamics, and some applications of the present treatment are found in Refs. [124,126,127,128].

#### 5.5.2. Statistical Redistribution of Tunneling Paths

In view of the nature of this review article, we briefly show an example of chaos that is induced by tunneling dynamics, just for curiosity. The Hamiltonian is
(96)$$H=\frac{1}{2m_{x}}p_{x}^{2}+\frac{1}{2m_{y}}p_{y}^{2}+\frac{1}{2}\left(x^{2}+y^{2}\right)+x^{2}y-\frac{1}{3}y^{3},$$
where the masses are chosen as $m_{x}=1.0087$ and $m_{y}=1.0$ so as to break the symmetry of $C_{3}$. The vibrational modes and the corresponding tori in the Poincaré surface of section are depicted in Figure 10. The bottom panel displays a long time Poincaré section arising from a single trajectory, which repeats dynamical tunneling from time to time among tori, thus giving rise to chaos effectively. Here, however, we have intensionally disregarded the damping of the amplitude induced by tunneling. Careful inspection will find the faint trace of a couple of tori [125]. See ref. [125] for details about the present tunneling dynamics.

## 6. Chaos Arising from Repeated Bifurcation and Merge of the Quantum
Wavepackets on Nonadiabatically Coupled Potential
Basins

One of the characteristic features of quantum chaos is reflected on the energy level statistics such as the patterns in the nearest neighbor level spacing distribution (NNLSD) [114,129]. The extensive level repulsions among the states are widely recognized to be the origin of the Wigner distribution in NNLSD. It is therefore natural to pay attention to the *nonadiabatic interactions* as one of the key factors to modulate the structure of the energy levels. As such, the theory of level dynamics [9,130,131,132] sets a focus on nonadiabatic interactions as a factor that can modulate the energy levels in an anomalous fashion. Shudo [133,134] explicitly clarified the effects of an avoided crossing on the level distributions, the avoided crossing of which is induced by a variation of a part (s) of interaction potential but not by those interactions presented in Equation (Equation 12). The nonadiabatic interactions of Equation (Equation 12) are basically about electronic-state mixing due to the *nuclear kinematic coupling* on the occasion of the variation of nuclear configuration (molecular geometry) and are one of the most critical interactions in chemical dynamics [[14],[16],[17],[18],[19],,[20]]. Therefore, the beautiful methods in the level dynamics does not seem to apply to the dynamics of molecules.

Besides, we are quite often concerned with the wavepacket dynamics of electrons rather than the electronic energy levels at a given nuclear configuration. In other words, the essential characteristics of molecular nonadiabatic dynamics is seen when the relevant quantum wavepackets *bifurcate* and coherently *merge* (remix) many times at the molecular geometry of significant nonadiabatic interactions. It is quite natural therefore to suspect that the wavepacket bifurcation and merging should be the dynamical origin of purely quantum chaos in place of “stretching and folding” in the baker’s transformation for classical chaos. We hence survey the nuclear wavepacket dynamics of bifurcation and remixing in this section first and then proceed to the nonadiabatic electron wavepacket dynamics in the next section.

### 6.1. Experimental Observation of Bifurcation and Merge of the Quantum Wavepackets

We here show that “bifurcation and merging” of quantum wavepackets can be observed with the modern real-time experiments. Among the cutting-age experimental methodologies to track the femtosecond-scale nuclear motions in chemical dynamics, the time- and angle-resolved photoelectron spectroscopy (TRPES) has been very well established since 1999 [135,136,137], when the group of this author happened to start the first complete ab initio full quantum mechanical simulation to track nuclear wavepackets in TRPES [138,139,140]. Among others, we showed for the first time that the instance of bifurcation and merging of nuclear wavepackets in passing across the nonadiabatic regions can be directly observed [141,142]. That is, such bifurcation and merging are of physical reality. Now other novel experimental methodologies are available.

For readers who might be interested in the recent experimental progress since then, which are related to observation nonadiabatic dynamics, we refer to the following works (we apologize for not a complete list). T. Suzuki is the first who used photoelectron imaging technique to observe the dynamics passing through an intersystem crossing of pyradine [143,144]. Wörner and his group observed the passage of NO${}_{2}$ molecule across the conical intersection by means of the high harmonic spectroscopy [145] and time-resolve photoelectron spectroscopy [146]. Further novel methods have been introduced, such as attosecond stimulated X-ray Raman spectroscopy [147], time-resolved fluorescent spectroscopy [148], ultrafast electron diffraction technique [149], and so on. We also found that the wavepacket bifurcation can be directly observed not only by pump-probe technique by means of pulse lasers but with induced photoemission spectroscopy from molecules placed under a CW laser [150].

The basic experimental setting for the time-resolved photoelectron spectroscopy is as follows: (1) First excite a ground state wavepacket to an electronically excited state by a pump pulse-laser. This laser is of a sufficiently short width so that the nuclear wavepacket thus excited begins to run spontaneously, which eventually passes across the nonadiabatic region(s). Then, (2) shine another laser (probe laser) to ionize the excited state. (3) Measure the energy and angular distributions of photoelectrons as a function of the time delay (the time lag between the probe and pump), and thereby retrieve the information of the dynamics of the nuclear (chemical) dynamics. This TRPES technique is quite successful in that there is no closed channel to be probed.

The theoretical framework of TRPES is outlined below [138,139,140]. Briefly, the total wavefunction is expanded in terms of the three relevant electronic states: $\Phi_{1}$ (with a potential function $V_{1}$), $\Phi_{2}$ (with $V_{2}$), and $\Phi_{k}^{\left(-\right)}$ (ion state), as
(97)$$\Psi \left(r,R,t\right)=\chi_{1}\left(R,t\right)\Phi_{1}\left(r;R\right)+\chi_{2}\left(R,t\right)\Phi_{2}\left(r;R\right)+\int dk\chi_{k}\left(R,t\right)\Phi_{k}^{\left(-\right)}\left(r;R\right),$$
where $\Phi_{k}^{\left(-\right)}\left(r;R\right)$ is a scattering state having the in-going boundary condition [49] with k labeling the wave vector of the photoelectron. The time-dependent coupled Schrödinger equations for the vibrational wavepackets $\chi_{1}$, $\chi_{2}$, and $\chi_{k}$ are to be solved in a coupled Schrödinger equations (note that $\chi_{k}$ is continuous with respect to k) under all the matrix elements being evaluated in full quantum mechanics [138,139,140]. For instance, we need the matrix elements with respect to the electronic continuum representing the electronic ionization process. This is not an easy task. Once they are all solved through, the photoelectron spectra are calculated from the vibrational wavepackets after the probe interaction as
(98)$$P\left(\varepsilon_{k}\right)=k\int dRd\hat{k}{\left|\chi_{k}\left(R\right)\right|}^{2}.$$

Figure 11 shows the photoelectron spectra of NaI molecule exhibiting the wavepacket bifurcation and merging, taken for the pump-probe delays of 675, 700, 725, and 750 fs. Snapshots of the absolute square of the wavepackets on the diabatic potentials are shown in the upper column. The red and blue ones represent ${\left|\chi_{1}^{\left(dia\right)}\left(R,t\right)\right|}^{2}$ on $V_{1}^{\left(dia\right)}\left(R\right)$ and ${\left|\chi_{2}^{\left(dia\right)}\left(R,t\right)\right|}^{2}$ on $V_{2}^{\left(dia\right)}\left(R\right)$, respectively. These wavepackets are passing across the crossing point (nonadiabatic region) from the right to the left. The photoelectron signals, the kinetic energy distribution of photoelectrons, at each time are correspondingly draw in the lower column. It is clearly seen that the peaks of photoelectron signals are modulated in real time from one to two and three peaks, highlighting the wavepacket bifurcation to those components. These dynamics can be comprehended primarily with the Condon principle: The wavepackets running on the lower potential energy make the higher photoelectron kinetic energy signals. The wavepacket bifurcation and merging and can thus be physically observed in real time.

### 6.2. Chaotic Eigenfunctions in Nonadiabatically Coupled Potential Functions

We next survey how the infinite repetition of bifurcation and merge of the quantum wavepackets results in the specific signature in eigenfunctions. Fujisaki systematically investigated this aspect by placing a focus on the transition from regular to chaotic states of the wavefunctions as functions of energy and the magnitude of nonadiabatic coupling elements [151,152]. The system used was the so-called Heller model, which was defined in their pioneering work [153]. It is defined as (see also Figure 12)
(99)$$\begin{array}{cc} H & =\left(\begin{array}{cc} T+V_{A} & J\\ J & T+V_{B} \end{array}\right)\\ T & =\frac{1}{2}\left(p_{x}^{2}+p_{y}^{2}\right),\;V_{A}=\frac{1}{2}\left(\omega_{x}^{2}x^{2}+\omega_{y}^{2}y^{2}\right)+\epsilon_{A},\;V_{B}=\frac{1}{2}\left(\omega_{x}^{2}\xi^{2}+\omega_{y}^{2}\eta^{2}\right)+\epsilon_{B} \end{array}$$
with ξ=(x+2asinθ)cos2θ+(y−2acosθ)sin2θ and η=−(x+2asinθ)sin2θ+(y−2acosθ)cos2θ, where *T* is a nuclear kinetic energy (nuclear masses are assumed to be unity), $V_{i}$ (i=A,B) two-dimensional diabatic harmonic potentials of each electronic state *i*, *J* the nonadiabatic coupling constant between the electronic states induces an interaction between the two otherwise uncoupled harmonic potentials. $\epsilon_{i}$(i=A,B) indicates the bottoms of each harmonic potential. The geometrical meanings of *a* and θ are shown in Figure 12. The angle 2θ referred to as the Duschinsky angle cause a coupling between *x* and *y*. Unlike the original Heller model, we chose the frame for $V_{A}$ to be ‘diagonalized’ (no crossing terms like xy in $V_{A}$). The location of the crossing manifold (usually called crossing seam) should be presented in the original paper [152].

Since the Heller model is simply an uncoupled pair of harmonic potentials at J=0, it can have merely regular states depending on the values on *J* and θ. Yet, the intense mixing induced by these coupling elements lead the wavefunctions and energy spectra to exhibit chaotic patterns. As an example, Figure 13 demonstrates the spatial distribution of some (selected) very chaotic wavefunctions. The chaoticity was quantified in terms of a transition from the Poisson to Wigner distributions in the nearest neighbor level spacing distribution (NNLSD), $\Delta_{3}$ statistics, and so on [151]. Here in the graph of Figure 13, only the so-called amplitude distributions of the eigenfunctions, proposed by Berry [154], on one of the surfaces (*A*) are shown. The definition of the amplitude distributions is
(100)$$P\left(\phi \right)=\frac{1}{\sqrt{2\pi \sigma^{2}}}\exp\left(-\frac{\phi^{2}}{2\sigma^{2}}\right),$$
where P(ϕ) is the frequency distribution of the amplitude of the eigenfunction that is randomly sampled at points that are energetically accessible in configuration space on each potential. About 6000 points are randomly sampled in Figure 13, and it turns out that P(ϕ) is found to be fitted with a large σ, reflecting the spatially wide distribution due to chaos. Although not shown here, we found that the regular wavefunctions give much smaller σ (far narrower distributions) [151,152]. Thus, we have seen a regular to chaotic transition of eigenfunctions (and eigenvalues as well, not shown here) in nonadiabatically coupled quantum system.

### 6.3. Need for Measures of Chaoticity in Quantum Wavepackets

One of the most interesting issues in quantum chaos is how to measure the extent of chaoticity. This would depend on the definition of quantum chaos [2,3,4,5,6,7,8,9,10,11,12]. Characterization of chaoticity embedded in discrete quantum *energy levels* has been extensively studied (see Refs. [5,9,11] for instance). For example, the measures such as NNLSD, $\Delta_{3}$ statistics, and so on, work well to capture chaos. On the other hand, we need to characterize chaoticity in the *dynamics* of wavefunctions of molecules. Berry’s amplitude distributions of the eigenfunctions P(ϕ) in Equation (Equation 100) can be used as one of such measures by applying at each time of the time-propagation of a packet. However, this measure seems to be based on a rather intuitive expectation that the chaotic wavefunction should be always widely distributed in space. Another concern is whether it can distinguish well between regular systems and those of quantum localization and the quantum scar.

The studies on classical chaos have already established such measures like the Lyapunov exponent [155], the real-complex valuedness of the eigenvalues of the stability matrix, KS entropy, and so on [1]. In practical studies on chemical dynamics we often need a crisp measure to detect and/or characterize chaoticity in the propagation of wavefunctions. However, it is known that quantum coherence working in quantum wavepackets particularly make a significant difference from the classical one. Suppose we have two wavepackets $\phi_{\mu }\left(t\right)$ and $\phi_{\mu^{\prime }}\left(t\right)$, which are slightly different from each other in their initial conditions μ and $\mu^{\prime }$, the initial condition of two wavepackets for instance. In marked contrast to the exponential deviation of two classical trajectories of slightly different initial conditions, the following overlap between the two packet
(101)$$O_{\mu \mu^{\prime }}\left(t\right)={\left|\langle \left.\phi_{\mu }\left(t\right)\right|\phi_{\mu^{\prime }}\left(t\right)\rangle \right|}^{2}$$
is kept constant as long as they are subject to the same Hamiltonian. For the same reason, quantities like the Shannon entropy
(102)$$S\left(t\right)=-\int dq{\left|\phi_{\mu }\left(q,t\right)\right|}^{2}\log{\left|\phi_{\mu }\left(q,t\right)\right|}^{2}$$
is conserved as well. Then, a primitive idea to break the coherence is as follows [156]. Prepare a projection operator
(103)$$P_{i}=\int_{A_{i}}dr|r\rangle \langle r|$$
where $A_{i}$ is a subset (local area) of the entire space and satisfies $A_{i}\cap A_{j}=\emptyset$ for i≠j. In place of the overlap integral $O_{\mu \mu^{\prime }}\left(t\right)$
(104)$$O_{\mu \mu^{\prime }}\left(t\right)={\left|\langle \phi_{\mu }\left(t\right)\left|\sum_{i}P_{i}\right|\phi_{\mu }\left(t\right)\rangle \right|}^{2}$$
we consider the following modulation
(105)$$D_{\mu \mu^{\prime }}\left(t\right)=\sum_{i}{\left|\langle \phi_{\mu }\left(t\right)\left|P_{i}\right|\phi_{\mu^{\prime }}\left(t\right)\rangle \right|}^{2},$$
which obviously destroys the coherence among the different regions. In chaos $\phi_{\mu }\left(t\right)$ and $\phi_{\mu^{\prime }}\left(t\right)$ deviate from each other in configuration space, and $D_{\mu \mu^{\prime }}\left(t\right)$ is anticipated to exhibit a quick damp from a slight difference in the initial condition [156]. A good compromise to balance the number of $A_{i}$ and size is necessary to sensitive yet economical detection of chaos. It has been shown numerically that the method works very well even for the nonadiabatic dynamics as in the above Heller model system [156]. However, this decoherence method costs much labor in many dimensional systems. We hence need general and tractable alternatives. We will be back to this aspect in the next section.

## 7. Chaos in Nonadiabatic Electron Dynamics of Molecules


In this section, we review chaotic dynamics in electronic state manifolds, those states involved in which are mixed frequently and extensively by the nuclear kinematic couplings, or as vaguely termed, nonadiabatic interactions [21,23,24,25,26,27]. (We do not review the traditional and canonical theories of nonadiabatic transitions for nuclear wavefunctions, essentially originating from the spirit of the Landau-Zener-Stuekelberg theory [14,16,17,19,20]).

### 7.1. The Hamiltonian Studied for Electron Dynamics

#### 7.1.1. Electron Dynamics

The coupled Equations (Equation 11) based on the Born-Huang expansion, Equation (Equation 9), set the theoretical foundation of molecular science. Yet, various drawbacks arise coming from the time-independent representation of the electronic wavefunctions $\left\{\Phi_{I}\left(r;R\right)\right\}$. First of all, it is technically hard to prepare a set of potential functions $\left\{V_{I}\left(R\right)\right\}$, on which the nuclear wavepackets $\left\{\chi_{I}\left(R,t\right)\right\}$ propagate in time. In particular, the most serious situation appears when the electronic states are densely quasi-degenerate and are strongly coupled to one another through the nonadiabatic interactions. In such a case a huge time-dependent fluctuation over the electronic states can appear. To cope with those difficulties, we resume the dynamics with time-dependent electronic wavefunctions from the beginning.

We hereby restart our basic framework from the following representation of the total Hamiltonian operator [21,157]
(106)$$\begin{array}{c} H(R,\;\mathrm{elec}\;)\\ \equiv \frac{1}{2}\sum_{k}{\left({\hat{P}}_{k}-i\hbar \sum_{IJ}|\Phi_{I}\rangle X_{IJ}^{k}\langle \Phi_{J}|\right)}^{2}+\sum_{IJ}|\Phi_{I}\rangle H_{IJ}^{\left(\mathrm{el}\right)}\langle \Phi_{J}| \end{array}$$
where ${\hat{P}}_{k}$ is the nuclear momentum operator. R indicates the nuclear coordinates, and we represent the electronic states in the Hilbert space, not in the configuration space r. Note that ${\hat{P}}_{k}$ are to be operated only on the nuclear wavefunctions, not on the electronic wavefunctions $\Phi_{I}\left(r;R\right)\;$, since such kinematic interactions have already been taken into account in $X_{IJ}^{k}$ of Equation (Equation 106). The sum over the electronic states suffixed with *I* and *J* may include continuum states.

No approximation has been made so far, and, we here introduce an approximation that the nuclear momentum operators ${\hat{P}}_{k}$ are replaced with their classical counterparts $P_{k}$ as
(107)$$\begin{array}{c} \tilde{H}\left(R,P,\;\mathrm{elec}\;\right)\\ \equiv \frac{1}{2}\sum_{k}{\left(P_{k}-i\hbar \sum_{IJ}|\Phi_{I}\rangle X_{IJ}^{k}\langle \Phi_{J}|\right)}^{2}+\sum_{IJ}|\Phi_{I}\rangle H_{IJ}^{\left(\mathrm{el}\right)}\langle \Phi_{J}| \end{array}$$

Then the total wavefunction subject to the Hamiltonian of Equation (Equation 106) is to be reduced to an electron wavepacket state running on paths R(t) (but not necessarily Newtonian trajectories). More explicitly, we expand it as
(108)$$|\Phi_{\mathrm{elec}}\left(t;R\left(t\right)\right)\rangle =\sum_{I}C_{I}\left(t\right)|\Phi_{I}\left(r;R\right)\rangle {|}_{R=R(t)},$$
along each R(t), where $\left\{\Phi_{I}\left(r;R\right)\right\}$ is a basis functions taken at each nuclear coordinate R, and $\Phi_{I}\left(r;R\right)$ do not have to be the eigenfunctions of the electronic Hamiltonian. Equations (Equation 107) and (Equation 108) are widely referred to as mixed quantum-classical representation. The time-dependent coefficients $C_{I}\left(t\right)$ are to be evaluated with the coupled equations of motion for electron wavepackets in such a way that
(109)$$i\hbar \frac{dC_{I}}{dt}=\sum_{J}\left[H_{IJ}^{\left(\mathrm{el}\right)}-i\hbar \sum_{k}{\dot{R}}_{k}X_{IJ}^{k}-\frac{\hbar^{2}}{4}\sum_{k}\left(Y_{IJ}^{k}+Y_{JI}^{k*}\right)\right]C_{J},$$
where $H_{IJ}^{\left(\mathrm{el}\right)},$$X_{IJ}^{k},$ and $Y_{IJ}^{k}$ have been defined in Equations (Equation 12) and (Equation 13), respectively. Again, the electronic basis $\left\{\Phi_{I}\left(r;R\right)\right\}$ are not demanded to be adiabatic ones. The bra-ket notation used here indicates integration over the electronic coordinates r only. Furthermore, the second order derivative terms $Y_{IJ}^{k}$ in Equation (Equation 109) are quite often neglected in practice because they are always accompanied by the small quantity $\hbar^{2}$ [24,25], although it is not always negligible in general.

#### 7.1.2. Nuclear Dynamics: Path Branching

The nuclear paths R(t) are driven by the force field provided by the electronic states, more explicitly in terms of the force matrix $F_{IJ}^{k}$ (see ref. [20]) defined as
(110)$$F_{IJ}^{k}=-\frac{\partial H_{IJ}^{\left(\mathrm{el}\right)}}{\partial R_{k}}-\sum_{K}\left(X_{IK}^{k}H_{KJ}^{\left(\mathrm{el}\right)}-H_{IK}^{\left(\mathrm{el}\right)}X_{KJ}^{k}\right)+i\hbar \sum_{l}{\dot{R}}_{l}\left[\frac{\partial X_{IJ}^{l}}{\partial R_{k}}-\frac{\partial X_{IJ}^{k}}{\partial R_{l}}\right],$$
which arises from the variational principle or the Hamilton canonical equations of motion for (R,P) [20].

The non-zero off-diagonal elements of $F_{IJ}^{k}$ can induce branching of the nuclear paths R(t) [21]. If, on the other hand, the force matrix is diagonal, that is, $F_{IJ}^{k}=0$ for I≠J, each of $F_{II}^{k}$ gives a Newtonian force on each potential function *I*, which gives the theoretical foundation for the so-called ab initio dynamics, molecular dynamics, the first principle dynamics, and so on.

#### 7.1.3. Nuclear Dynamics: Mean-Field Approximation (Semiclassical
Ehrenfest Theory)

To save the work needed for the path branching calculations [22], we often adopt a mean-field approximation, the so-called semiclassical Ehrenfest theory (SET), in which the force matrix of Equation (Equation 110) is to be averaged over the electron wavepacket as
(111)$$\begin{array}{c} \tilde{H}\left(R,P,\;\mathrm{elec}\;\right)\\ \equiv \frac{1}{2}\sum_{k}{\left(P_{k}-i\hbar \sum_{IJ}|\Phi_{I}\rangle X_{IJ}^{k}\langle \Phi_{J}|\right)}^{2}+\sum_{IJ}|\Phi_{I}\rangle H_{IJ}^{\left(\mathrm{el}\right)}\langle \Phi_{J}| \end{array}$$
which gives rise to a single force for a single path. If the basis set $\left\{\Phi_{I}\left(r;R\right)\right\}$ happens to be complete, Equation (Equation 111) is reduced to the form of Hellmann–Feynman force
(112)$$\langle F_{k}\rangle =-\langle \Psi_{\mathrm{elec}}\left|\frac{\partial {\hat{H}}^{\left(\mathrm{el}\right)}}{\partial R_{k}}\right|\Psi_{\mathrm{elec}}\rangle .$$

The use of the mean-field approximation is justified for short-time propagation, typically shorter than 100 fs. Indeed, it has been numerically shown [21] that the electronic-state mixing is very well (accurately) reproduced by SET for a short time, provided that the nonadiabatic coupling elements $X_{IK}^{k}$ and electronic matrix elements $H_{IK}^{\left(\mathrm{el}\right)}$ are both correctly evaluated. As the nuclear positions proceed to some extent, the Ehrenfest paths should be modified accordingly [22].

### 7.2. Longuet–Higgins (Berry) Phase and Lorentz-like Force

Before proceeding to the electronic-state chaos, we make a detour to a specific force working on the nuclei, which can cause breaking the mirror symmetry of molecular shape, namely molecular chirality.

Let us review the molecular Hamiltonian of Equation (Equation 107) now in classical electromagnetic field
(113)$$\begin{array}{c} \tilde{H}\left(R,P,\;\mathrm{elec}\;,A\right)\equiv \\ \frac{1}{2}\sum_{k}{\left(P_{k}-\frac{Z_{k}e}{c}A_{k}-i\hbar \sum_{IJ}|\Phi_{I}\rangle X_{IJ}^{k}\langle \Phi_{J}|\right)}^{2}+\sum_{IJ}|\Phi_{I}\rangle H_{IJ}^{\left(el\right)}\left(A^{ele}\right)\langle \Phi_{J}| \end{array}$$
where $A_{k}\left(R\right)$ are the (classical) electromagnetic vector potentials on nuclei. The electronic Hamiltonian also includes the vector potential $A^{ele}$. In this expression we notice that the nonadiabatic couplings mechanically stand parallel with the vector field A, which suggests the qualitative roles of the nonadiabatic couplings working on the nuclear momentum may be akin to those of the vector potentials.

Indeed it is well-known that there is a mathematical similarity between the Aharonov–Bohm phase [158] and the Longuet–Higgins phase [159], the latter being well known in chemical dynamics relevant to the so-called conical intersection, which is a special case of nonadiabatic interaction. (see ref. [17] for the interesting properties of conical intersection.) These two phases have been unified as the so-called geometrical phase, often called Berry’s phase [160,161]. In light of the above parallelism in Equation (Equation 113), we suspect another similarity to exist between them: As the electromagnetic Lorentz force works on a charged particle running in a magnetic field, a novel force, which we call the Lorentz-like nonadiabatic force, appears to work on nuclei running in a field created by the nonadiabatic interaction field [21,162].

To be more specific, let us imagine a molecular system as illustrated in Figure 14.

At the center we have an atom C, say carbon atom, and around it lie three atomic or molecular elements A${}_{1}$, A${}_{2}$, A${}_{3}$. All the elements A${}_{1}$, A${}_{2}$, A${}_{3}$ in the molecule CA${}_{1}$A${}_{2}$A${}_{3}$ are mutually different and sit stably on a common plane. Suppose an external atom A is coming in on the same plane of an x−y coordinates. (A${}_{1}$, A${}_{2}$, A${}_{3}$, and A should be all different species from each other.) The *z*-direction is set to be perpendicular to this plane and the plus or minus of this direction determines the right or left with respect to the plane. Let $\left(x_{a},y_{a},z_{a}\right)$ be a point of the incident atom A. Then the force matrix in $z_{a}$-direction (perpendicular to the plane) [162] is given such that
(114)$$\begin{array}{cc} F_{IJ}^{z_{a}}= & -{\left(\frac{\partial H^{\left(el\right)}}{\partial R_{z_{a}}}\right)}_{IJ}-i\hbar \left({\dot{R}}_{x_{a}}\left[\frac{\partial X_{IJ}^{x_{a}}}{\partial R_{z_{a}}}-\frac{\partial X_{IJ}^{z_{a}}}{\partial R_{x_{a}}}\right]+{\dot{R}}_{y_{a}}\left[\frac{\partial X_{IJ}^{y_{a}}}{\partial R_{z_{a}}}-\frac{\partial X_{IJ}^{z_{a}}}{\partial R_{y_{a}}}\right]\right)\\ & -i\hbar \sum_{b\neq a}^{A_{1},\;A_{2},\;A_{3}}\sum_{j_{b}}^{x,y,z}{\dot{R}}_{j_{b}}\left[\frac{\partial X_{IJ}^{j_{b}}}{\partial R_{z_{a}}}-\frac{\partial X_{IJ}^{z_{a}}}{\partial R_{j_{b}}}\right] \end{array}$$
where the suffix *b* specifies A${}_{1}$, A${}_{2}$, and A${}_{3}$, and $j_{b}$ indicates the $\left(x,y,z\right)$-coordinate of each element. This is a force while the state undergoes nonadiabatic transition from an *I*th to the *J*th one. Just for simplicity, we assume that all the atoms other than the incident atom A stay still, thus neglecting the possible vibrational motion. For the sake of clear presentation, we simply set to
(115)$${\dot{R}}_{j_{b}}=0\;\;\;\;\;\;\;\;\;\;\left(j=x,y,z\;\;\mathrm{and}\;\;b\neq a\right).$$

Hence only the atom A is supposed to be moving on the plane with a nonzero value of $\left({\dot{R}}_{x_{a}},{\dot{R}}_{y_{a}}\right)$ and ${\dot{R}}_{z_{a}}=0$. $F_{IJ}^{z_{a}}$ is then reduced to
(116)$$F_{IJ}^{z_{a}}=-{\left(\frac{\partial H^{\left(el\right)}}{\partial R_{z_{a}}}\right)}_{IJ}-i\hbar \left({\dot{R}}_{x_{a}}\left[\frac{\partial X_{IJ}^{x_{a}}}{\partial R_{z_{a}}}-\frac{\partial X_{IJ}^{z_{a}}}{\partial R_{x_{a}}}\right]+{\dot{R}}_{y_{a}}\left[\frac{\partial X_{IJ}^{y_{a}}}{\partial R_{z_{a}}}-\frac{\partial X_{IJ}^{z_{a}}}{\partial R_{y_{a}}}\right]\right).$$

We here define the Lorentz-like nonadiabatic force with the second term of the right hand side of this equation as [162]
(117)$$f_{IJ}^{z_{a}}\equiv -i\hbar \left({\dot{R}}_{x_{a}}\left[\frac{\partial X_{IJ}^{x_{a}}}{\partial R_{z_{a}}}-\frac{\partial X_{IJ}^{z_{a}}}{\partial R_{x_{a}}}\right]+{\dot{R}}_{y_{a}}\left[\frac{\partial X_{IJ}^{y_{a}}}{\partial R_{z_{a}}}-\frac{\partial X_{IJ}^{z_{a}}}{\partial R_{y_{a}}}\right]\right)$$
and simply write as
(118)$$F_{IJ}^{z_{a}}=-{\left(\partial_{z_{a}}H^{\left(el\right)}\right)}_{IJ}+f_{IJ}^{z_{a}},$$
where $-{\left(\partial_{z_{a}}H^{\left(el\right)}\right)}_{IJ}$ is the short-hand notation of the off-diagonal Hellmann-Feynman term. (Note that the diagonal elements of the nonadiabatic interaction $X_{II}$ are zero if the function used as the basis functions are real. However, this is not the case for complex-valued wavefunctions.)

The expression in Equation (Equation 117) can be reexpressed in a more familiar manner: Let us consider a vector of the force matrix elements as
(119)$${\vec{f}}_{IJ}^{a}=\left(f_{IJ}^{x_{a}},f_{IJ}^{y_{a}},f_{IJ}^{z_{a}}\right).$$

In terms of the vector ${\vec{X}}_{IJ}^{{}_{a}}\left(R\right)$ of the nonadiabatic interaction fields
(120)$${\vec{X}}_{IJ}^{{}_{a}}\left(R\right)=\left(X_{IJ}^{x_{a}}\left(R\right),X_{IJ}^{y_{a}}\left(R\right),X_{IJ}^{z_{a}}\left(R\right)\right)$$
we define a vector field
(121)$${\vec{Z}}_{IJ}^{{}_{a}}\left(R\right)=-i\hbar {\vec{\nabla }}^{a}\times {\vec{X}}_{IJ}^{{}_{a}}\left(R\right),$$
in which the dependence on the nuclear coordinates R is explicitly denoted. Then ${\vec{f}}_{IJ}^{a}$ is expressed as
(122)$${\vec{f}}_{IJ}^{a}\left(R\right)={\vec{v}}^{a}\times {\vec{Z}}_{IJ}^{{}_{a}}\left(R\right),$$
where ${\vec{v}}^{a}$ is the velocity vector of the atom A, and ${\vec{\nabla }}^{a}\times$ indicates the curl on the coordinates for the atom A only [162].

It is immediately noticed that this expression is essentially of the same mathematical form as that behind the Fleming *left-handed* rule in classical electromagnetic theory. The Lorentz force distinguishes the front and back sides of a plane made up by an electron current and a magnetic field. See Figure 15. Thus, a simple relationship is found between the electromagnetic Lorentz force and the Lorentz-like nonadiabatic force in such a way that
(123)$${\vec{A}}^{a}↔{\vec{X}}_{IJ}^{{}_{a}},\;\;\;\;{\vec{B}}^{a}↔{\vec{Z}}_{IJ}^{{}_{a}},\;\;\;\;{\dot{R}}_{a}\times q_{a}eB_{a}\;↔{\vec{v}}^{a}\times {\vec{Z}}_{IJ}^{{}_{a}},$$
where B is a magnetic field.

We thus recognize that the nonadiabatic force can distinguish right and left with respect to the plane of the vector field of nonadiabatic interactions, which can serve as one of the fundamental mechanisms of the origin of molecular chirality. Yet, further study is needed to quantify the actual magnitude of anisotropic force.

### 7.3. Chaotic Electron Dynamics in Densely Quasi-Degenerate
Electronic-State Manifold; B${}_{12}$ Cluster as an Example

As a candidate among molecules possibly having electronic-state chaos, we study some of the highly excited states of a cluster consisting of 12 boron atoms, B${}_{12}$. The nonadiabatic interactions working among the excited states of this cluster cause very frequent transitions among adiabatic states [163,164,165], resulting in frequent and continual state-mixing as represented as (recall Equation (Equation 108))
(124)$$\sum_{I}^{\mathrm{adiabatic}}C_{I}\left(t\right)\Phi_{I}^{\mathrm{ad}}\left(r;R\left(t\right)\right).$$

In contrast to classical chaos, there is no crisp criterion with which to define or identify chaos in quantum dynamics, to the best of our knowledge. Below we only quantify the extent of chaoticity of the relevant electron dynamics represented as in Equation (Equation 124). A special attention is paid to the random mixing among the deterministic electronic states in the state space. We then discuss the possible manifestation of those strong chaoticity in chemical and physical properties of the clusters.

#### 7.3.1. Nearest Neighbor Level Spacing Distribution (NNLSD)

At any selected time t, or at any molecular shape R, one can diagonalize the electronic Hamiltonian matrix $H_{IJ}^{\left(\mathrm{el}\right)}$ in Equation (Equation 109). The energy levels are therefore geometry dependent. We have surveyed the NNLSD for the B${}_{12}$ cluster (see ref. [164] and its supplemental material stored in J. Chem. Phys.). It turns out that whereas the NNLSD shows a clear Wigner distribution in case of low symmetry of molecular shape, level clustering occurs to some extent when the molecule is of higher symmetry, actually C${}_{3v}$ symmetry. This is natural because the systems of higher symmetry tend to bear accidental degeneracy more often. On the other hand, the possibility (measure) for this system to take C${}_{3v}$ symmetry is actually zero in the dynamical studies of variable R(t). Thus, we can judge that the present system has a chaotic structure in its energy level. However, our goal is to clarify and characterize the chaotic feature possibly hidden in the individual electronic wavepackets.

#### 7.3.2. Diffusive Dynamics in the State Space: Fractional Brown Motion

It is experimentally hard to pick and prepare a pure adiabatic excited state out of the densely quasi-degenerate electronic state. Even if one could successfully pump the ground state to one of the highly adiabatic states, the resultant state would quickly begin to undergo continual nonadiabatic mixing with other states, each of which in turn further mixes with other states. These enduring state-mixings look like a diffusion as a whole in the electronic Hilbert space. Figure 16 shows the magnitude of ${\left|C_{I}\left(t\right)\right|}^{2}$ of Equation (Equation 124) in a color code in the (I,t) space. The brighter color indicates the larger ${\left|C_{I}\left(t\right)\right|}^{2}$. It visibly demonstrates that diffusive dynamics has been certainly realized in the electronic-state space.

To quantify the extent of the diffusive motion of the wavepackets in the electronic state-space explicitly, we survey the scaling property of the dynamics in terms of the fractional Brownian motion (fBm) of Mandelbrot [166]. Consider
(125)$$\sum_{J=1}{\left(J-I\left(0\right)\right)}^{2}{\left|C_{J}\left(t\right)\right|}^{2}=D_{I(0)}t^{\alpha },$$
where I(0) indicates the state number of the initial state (that is $C_{I}\left(0\right)$), and $C_{J}\left(t\right)$ is the mixing coefficient at time *t* starting from this initial I(0) state. Again $C_{J}\left(t\right)$ have been evaluated with the SET electron wavepackets. Recall that the exponent α=1.0 implies the ordinary diffusion and $D_{I(0)}$ is just the ordinary diffusion constant. States of the initial adiabatic state I(0)=3,40,100,300, and 500 have been numerically examined [164]. Furthermore, it turns out that the nonadiabatic electron dynamics of I(0)≥ 40 are indeed the fractional Brownian motion (fBm) in the sense of Equation (Equation 125). The exponent of the present fBm is found to be about α=0.15, which suggests a milder stochastic motion than ordinary diffusion. Interestingly, the values of the exponent is mostly independent of the initial states chosen, whereas the diffusion constants $D_{I(0)}$ significantly depend on the initial states, with the higher states having the larger diffusion constant.

Thus, Figure 16 convinces enough that the concept of adiabatic state in the Born-Oppenheimer approximation and the associated the energy level (potential energy hypersurface) all lose sense in the present situation. Besides, the dynamical aspect of chaos is far more directly relevant to the present huge electronic-state fluctuation than the static signatures of chaos like the level statistics.

#### 7.3.3. Shannon Entropy

We next calculate the Shannon entropy in terms of ${\left|C_{I}\left(t\right)\right|}^{2}$ taken from Equation (Equation 124), which is the probability for the electronic wavepacket to occupy the *I*th adiabatic state at time *t*. Hence,
(126)$$S=-\sum_{I}{\left|C_{I}\left(t\right)\right|}^{2}\log{\left|C_{I}\left(t\right)\right|}^{2}$$
quantifies the information about randomness in the electronic Hilbert space. Note this entropy is different from Equation (Equation 102). Incidentally, the present system is too large to apply the spirit of decoherence as in Equation (Equation 105). Figure 17 shows *S* for the three wavepackets, those starting from the adiabatic state #100, #200, and #300. A very quick rise from S=0 is observed for all the wavepackets up to about 5 fs, and much slower uprises follow since then. The absolute magnitude of the entropy is found to be larger for the higher state, which seems natural. In case of the adiabatic dynamics, where no electronic-state mixing is induced, the entropy is kept zero since ${\left|C_{I}\left(t\right)\right|}^{2}=1.0$. We thus see that the present electron dynamics keep distributing a large amount of the population over the extensive range in the Hilbert space.

#### 7.3.4. Lyapunov Exponents for the Loss of Electronic State Memory

In the present nonadiabatic electron wavepacket dynamics, the main origin of chaoticity is not nonlinearity in the nuclear path dynamics but the nonadiabatic electronic state-mixing. To quantify the chaoticity due to the nonadiabatic interactions directly, we compare two dynamics that start from the same initial condition and one running on an *adiabatic potential* and the other to run on the *nonadiabatic averaged potential*. Suppose we have the *I*th pure adiabatic state $\Phi_{I}^{ad}\left(r;R\left(t\right)\right)$ at the nuclear position R(t) at time *t*. Let a classical path run on this potential surface $V_{I}\left(R\right)$ for a short time Δt with an arbitrary momentum P(t), reaching a point at $R^{ad}\left(t+\Delta t\right)$ with the electronic state $\Phi_{I}^{ad}\left(r;R^{ad}\left(t+\Delta t\right)\right)$. Likewise we propagate a nonadiabatic electronic state starting from the same set of $\left(\Phi_{I}^{ad}\left(r;R\left(t\right)\right),R\left(t\right),P\left(t\right)\right)$ but the nonadiabatic interactions are taken account this time. The nuclear path then arrives at $R^{SET}\left(t+\Delta t\right)$, which is slightly different from $R^{ad}\left(t+\Delta t\right),$ but a significant different electronic wavepacket $\Phi_{\left(I\right)}^{SET}\left(r;R^{SET}\left(t+\Delta t\right)\right)$ is to be attained. The process is summarized as
(127)$$\left\{\begin{array}{c} \Phi_{I}^{ad}\left(r;R\left(t\right)\right)\to \Phi_{I}^{ad}\left(r;R^{ad}\left(t+\Delta t\right)\right)\;\;\;\;\;\;\;\;\mathrm{by}\;\mathrm{adiabatic}\;\mathrm{path}\;\;\;\;\;\;\;\;\;\\ \Phi_{I}^{ad}\left(r;R\left(t\right)\right)\to \Phi_{\left(I\right)}^{SET}\left(r;R^{SET}\left(t+\Delta t\right)\right)\;\;\;\mathrm{by}\;\mathrm{nonadiabatic}\;\mathrm{path}. \end{array}\right.$$

The deviation of $\Phi_{\left(I\right)}^{SET}\left(r;R^{SET}\left(t+\Delta t\right)\right)$ from $\Phi_{I}^{ad}\left(r;R^{ad}\left(t+\Delta t\right)\right)$ should reflect a memory loss of the initial state, and we quantify this the process by regarding it as an exponential decay of
(128)$$\begin{array}{c} {\left|\langle \Phi_{I}^{ad}\left(r;R^{ad}\left(t+\Delta t\right)\right)|\Phi_{\left(I\right)}^{SET}\left(r;R^{SET}\left(t+\Delta t\right)\right)\rangle \right|}^{2}\\ ={\left|\langle \Phi_{I}^{ad}\left(r;R\left(t\right)\right)|\Phi_{I}^{ad}\left(r;R\left(t\right)\right)\rangle \right|}^{2}\exp\left(-\xi_{I}\Delta t\right) \end{array}$$
where *t* is chosen arbitrary. Unfortunately the integral in the left hand side of Equation (Equation 128) is not easy to estimate. However, for a sufficiently short Δt, the following approximation holds very accurately
(129)$$R^{SET}\left(t+\Delta t\right)≃R^{ad}\left(t+\Delta t\right),$$
since their deviation is far slower and smaller. In fact, the deviation is numerically found to be about 0.005 Å in 1.0 fs (not shown graphically). Thus, we may define
(130)$$\xi_{I}\left(t;\Delta t\right)=-\frac{1}{\Delta t}\log{\left|\langle \left.\Phi_{I}^{ad}\left(r;R^{SET}\left(t+\Delta t\right)\right)\right|\Phi_{\left(I\right)}^{SET}\left(r;R^{SET}\left(t+\Delta t\right)\right)\rangle \right|}^{2},$$
and with use of Equation (Equation 124) it is estimated by
(131)$$\xi_{I}=-\frac{1}{\Delta t}\log{\left|C_{I}\left(t+\Delta t\right)\right|}^{2}.$$

Figure 18 represents the process of the above memory-loss process as observed in the selected states. Panel (a) shows that the nonadiabatic electronic state starting from the adiabatic #300 one quickly loses its memory about which adiabatic state it emerged from [167]. Actually it takes only 0.5 fs for the nonadiabatic state to lose most of the memory in ${\left|C_{I}\left(t\right)\right|}^{2}.$ The relevant exponents defined in Equation (Equation 128) is shown in panel (b). In this calculation, Δt was set to 0.5 fs and the time *t* was scanned. Panel (b) displays two exponents $\xi_{I}$ of Equation (Equation 131) for the nonadiabatic states starting from the #100 and #300 adiabatic states, respectively. They fluctuate from time to time with the average value around 5.0 fs${}^{-1}$. This implies that the adiabatic path can immediately lose its memory as soon as the nonadiabatic interactions are switched on, no matter where the paths run. In other words, the adiabatic states are entirely embedded in a wide chaotic sea.

#### 7.3.5. Turbulent Electron Flow in the Cluster

To observe one of the physical consequences from the present stochastic electron dynamics, we next survey the internal electron current driven by the nonadiabatic interactions. The probability current density of electrons $\vec{j}\left(r,t\right)$, which is also termed electron flux, quantifies and visualizes the flow pattern. $\vec{j}\left(r,t\right)$ is supposed to satisfy the continuity equation
(132)$$\frac{\partial \rho (r,t)}{\partial t}+\nabla \vec{j}\left(r,t\right)=0,$$
and in quantum mechanics $\vec{j}\left(r,t\right)$ is defined as
(133)$$\vec{j}\left(r,t\right)=\frac{\hbar }{2im_{e}}\left[\psi^{*}\nabla \psi -\psi \nabla \psi^{*}\right],$$
where $m_{e}$ and ψ are the mass and wave function of the involved particles [168]. For many-electron systems it is reduced to
(134)$$\begin{array}{cc} \vec{j}\left(r,t\right) & ={\left.\frac{\hbar }{2im_{e}}\left[\nabla_{r}\rho \left(r^{\prime },r\right)-\nabla_{r^{\prime }}\rho \left(r^{\prime },r\right)\right]\right|}_{r^{\prime }\to r}\\ & ={\left.\frac{\hbar }{2im_{e}}\sum_{i}^{\mathrm{NO}}n_{i}\left[\lambda_{i}^{*}\left(r^{\prime }\right)\nabla_{r}\lambda_{i}\left(r\right)-\nabla_{r^{\prime }}\lambda_{i}^{*}\left(r^{\prime }\right)\lambda_{i}\left(r\right)\right]\right|}_{r^{\prime }\to r}, \end{array}$$
where $\nabla_{r}$ and $\nabla_{r^{\prime }}$ are the nabla with respect to r and $r^{\prime }$, respectively, and $\left\{\lambda_{i}\right\}$ are the natural orbitals that are eigenfunction of the density operator $\hat{\rho }\left(t\right)$ ($\rho \left(r^{\prime },r\right)=\langle r^{\prime }\left|\hat{\rho }\right|r\rangle$) with occupation numbers $n_{i}$. Note that only the complex-valued natural orbitals can make contributions to Equation (Equation 134). Stationary-state electronic wavefunctions like most of the eigenfunctions of the electronic Hamiltonian ${\hat{H}}^{\left(el\right)}$ are real-valued (with exceptions with respect to angular momentum eigenfunctions) and thereby give zero flux only (see [169,170] for the general electronic and nuclear flux in general molecules, refs. [171,172,173,174] for the early applications of electron flux in chemical dynamics, ref. [175] for an extensive review, and ref. [176] for one of the latest).

Figure 19 shows the electron flux induced by the electron wavepacket dynamics starting from the 74th and 75th states, individually, with no nuclear kinetic energy but released spontaneously from the initial geometry. The snapshots have been taken at two very short times at 0.6 fs and 1.0 fs. These flux vectors are viewed from the top. The packet starting from the 74th state happens to exhibit the electron current in the counterclockwise direction at 0.6 fs, while the 75th counterpart shows the current mostly in the direction clockwise. Although the cluster seems to be in C${}_{3v}$ symmetry, it is relaxed to the lower one, and consequently, these circular patters are not conserved and shortly transformed to a more complicated flow. Indeed, only shortly after 1.0 fs both fluxes begin to fall into very complicated in spatial patterns. After all, those electron currents become turbulent or random flow (not shown graphically). The most exciting chemical study is then to identify the consequences of electronic turbulence in chemical reactivity and molecular properties.

Incidentally, one can define the flux of electronic energy [177], which is of course different from the above electron flux. It is quite interesting to track real-time energy flow in molecules. We have observed the turbulent flux of the electronic energy too in the above excited-state cluster dynamics [177].

We have thus observed the clear signatures of quantum chaos in molecular nonadiabatic electron dynamics.

### 7.4. The Long Life-Time of Dynamical Chemical Bonding: Hyper Resonance

The above diffusive and chaotic phenomena due to the very frequent nonadiabatic interactions prevent molecular dissociation and bring about a notion of quantum resonance we have not seen before. As resonance phenomena the Pauling resonance in the theory of chemical electronic structure [178] and the Feshbach resonance [49] in quantum scattering theory [50] are most relevant to the present study. The Pauling resonance in the so-called valence bond theory is regarded as a configuration mixing (interaction) among the so-called resonance structures, each of which corresponds intuitively to a valence bonding. The Feshbach resonance, or core-excited resonance, is schematically represented as
(135)$$M+A\to {\left(MA\right)}^{*}$$
dissociations eventually with a long lifetime,
where a state of a target nuclei, atom, or molecule (denoted as *M*) is pumped to the higher levels by borrowing the kinetic energy of an incident particle *A*. The Feshbach resonances and their overlapping set the foundation of Intramolecular Vibrational energy Redistribution (IVR) in the unimolecular dissociation dynamics, as discussed in Section 3.2. Thus, the dissociation channels are temporarily closed by a temporal compound state ${\left(MA\right)}^{*}$ surviving a long life-time. Both the Pauling and Feshbach resonances can be regarded basically as events on a single adiabatic potential surface.

During the continual state-mixing in the diffusive motion in B${}_{12}$, the electronic and nuclear energies are mutually exchanged through the nonadiabatic interactions. This phenomenon can be regarded as another kind of quantum mechanical resonance. The present dynamics found in B${}_{12}$ may be termed as *hyper-resonance* (resonance among the resonant states), since each adiabatic state is already in a Feshbach and/or Pauling resonance state. We refer to such a very long life-time tentative chemical bonding as *dynamical bonding* [165].

### 7.5. Intra-Molecular Nonadiabatic Electronic Energy Redistribution

#### 7.5.1. Huge Inflation of Phase-Space Volume

Despite the very high energy of those excited-state boron clusters, they do not readily break apart, since the dissociation channels of them tend to be closed by very frequent nonadiabatic transitions. This is a natural consequence of hyper-resonance. We then survey a possible mathematical ground of the long life-time in terms of a drastically simplified statistical theory based on the phase space theory, originally proposed by J. C. Light [179]. This theory has been extensively applied to and modified before in the context of cluster dynamics [180,181,182], Calvo [183,184,185,186,187], and Fujii [65,66].

We consider only the simplest dissociation of B${}_{12}$ cluster in which only one atom is evaporated from *N*-body cluster (N=12). The phase space theory under no external field predicts the dissociation rate *k* as
(136)$$k=\frac{W_{N-1}\left(E\right)}{\Omega_{N}\left(E\right)},$$
where $\Omega_{N}\left(E\right)$ is the density of the states of the *N*-particle cluster at the total energy *E*, and $W_{N-1}\left(E\right)$ denotes the total flux induced by the dissociation of a single atom [179]. As for a single adiabatic potential energy surface, say $V_{I}\left(R\right)$ at a nuclear configuration R, $\Omega_{N}\left(E\right)$ is approximately estimated with the expression
(137)$$\Omega_{N}^{\left(I\right)}\left(E\right)=\hbar^{-3N}\int dRdP\;\delta \left(E-H_{I}\left(R,P\right)\right),$$
where
(138)$$H_{I}\left(R,P\right)=\frac{P\cdot P}{2}+V_{I}\left(R\right),$$
with the obvious notation of the total nuclear momentum P. We assume that the total energy is high above the potential barrier of the dissociation channels and that there are not significant centrifugal barriers for the present non-rotating clusters. In such systems, the so-called bottle-neck volume, say $\partial \Omega_{N}^{\left(I\right)},$ of the phase-space volume $\Omega_{N}^{\left(I\right)}$ to the dissociation channel cannot be appropriately defined. In place of resorting to $\partial \Omega_{N}^{\left(I\right)}$ therefore, we use the phase space volume of the daughter cluster of N−1 particles, denoted as $\Omega_{N-1}^{\left(I\right)}\left(E-\Delta E\right)$, where ΔE is the energy for the dissociating particle to take away with the relative velocity $\vec{v}$. The flux is approximately expressed as
(139)$$W_{N-1}^{\left(I\right)}\left(E\right)=\int \Omega_{N-1}^{\left(I\right)}\left(E-\Delta E\right)\vec{v}\left(\Delta E\right)\cdot d\vec{v}.$$

(The reduced mass is set to be unity for simplicity.) Thus, the critical quantity for the statistical rate of the dissociation probability turns out to be the ratio
(140)$$\frac{\Omega_{N-1}^{\left(I\right)}\left(E-\Delta E\right)}{\Omega_{N}^{\left(I\right)}\left(E\right)}$$
for a given ΔE.

We further suppose that the phase space of a system under study has the mixing property and is completely statistical due to chaos. Then we may regard the quantity
(141)$$\Omega_{N}^{\left(I\right)}\left(E\right)-\Omega_{N-1}^{\left(I\right)}\left(E-\Delta E\right)$$
as a phase space volume for trajectories to “tentatively” remain bound as *N*-body cluster. Then the related entropy $s^{\left(I\right)}\left(E,\Delta E\right)$ can then be defined as
(142)$$s^{\left(I\right)}\left(E,\Delta E\right)=-\log\left(\frac{\Omega_{N}^{\left(I\right)}\left(E\right)-\Omega_{N-1}^{\left(I\right)}\left(E-\Delta E\right)}{\Delta \Omega }\right),$$
where ΔΩ represents the size of a unit cell of the phase space volume, such as $\hbar^{3N}$, which is used to count the number of cells in the volume $\left[\Omega_{N}^{\left(I\right)}\left(E\right)-\Omega_{N-1}^{\left(I\right)}\left(E-\Delta E\right)\right]$.

Although the state mixing is caused by the interaction
(143)$$-i\hbar \sum_{k=1}^{3N}{\dot{R}}_{k}\left(t\right)X_{IJ}^{k}\left(R\left(t\right)\right)$$
and/or $H_{IJ}^{\left(\mathrm{el}\right)}\left(R\left(t\right)\right)$ as in Equation (Equation 109), we herein assume *for simplicity* that the transition among the electronic states occur freely. Then the entropy confined in the individual state $s^{\left(I\right)}\left(E,\Delta E\right)$ should be represented by the sum of them as
(144)$$\begin{array}{c} s^{\left(\max\right)}\left(E,\Delta E\right)=\sum_{I}s^{\left(I\right)}\left(E,\Delta E\right)\\ =-\sum_{I}\log\left(\frac{\Omega_{N}^{\left(I\right)}\left(E\right)-\Omega_{N-1}^{\left(I\right)}\left(E-\Delta E\right)}{\Delta \Omega }\right) \end{array}$$

In such an extreme case the connected phase-space volume within ΔE is effectively extended and can be approximated as
(145)$$\prod_{I}\left[\Omega_{N}^{\left(I\right)}\left(E\right)-\Omega_{N-1}^{\left(I\right)}\left(E-\Delta E\right)\right].$$

On the other hand, the phase space volume directly connected to the dissociation channel should be represented as the simple sum of each as
(146)$$\sum_{I}\Omega_{N-1}^{\left(I\right)}\left(E-\Delta E\right).$$

Hence the the dissociation probability can be dramatically lowered from the order of Equation (Equation 140) to
(147)$$\frac{\sum_{I}\Omega_{N-1}^{\left(I\right)}\left(E-\Delta E\right)}{\prod_{J}\left[\Omega_{N}^{\left(J\right)}\left(E\right)-\Omega_{N-1}^{\left(J\right)}\left(E-\Delta E\right)\right]},$$
indicating that the relative phase-space volume towards the dissociation is by far smaller than the phase space volume of the original *N*-body cluster, and the probability for the cluster to come across the dissociation channel is extremely small. Consequently, the life-time of the clusters is expected to be far longer than in case of no nonadiabatic transitions.

#### 7.5.2. Intra-Molecular Nonadiabatic Electronic Energy Redistribution
(INEER): Nonadiabatic Interaction to Close Dissociation Channels

Nonadiabatic electronic-state chaos thus inflates the volume of phase-space of nuclear motion of clusters to a very large extent. Due to this inflation, the molecular states lose chances to reach one of their dissociation channels. That is, a dissociating molecule can be brought back to the bound state by nonadiabatic interactions, which would dissociate otherwise. Here is an example of such an event [165]. In Figure 20, a B${}_{12}$ of initial hindered geometry is prepared, which is close to a dissociation channel. (Such dissociation channels can be readily found by colliding a B${}_{11}$ cluster and a B atom) In panel (a) is shown a trajectory for the 47th adiabatic state at this geometry to run on its adiabatic potential surface (no nonadiabatic interaction). After 110 fs of running, one of the atoms leaves the cluster leaving B${}_{11}$ cluster behind. Likewise, the same 47th adiabatic state in panel (b) begins to run from the same geometry with the same momentum but with the nonadiabatic interactions explicitly included in the electron dynamics. The trajectory clearly demonstrates that an atom being about to leave the cluster is brought back again to the B${}_{12}$, thus evidencing that the dissociation channel is closed. This phenomenon is what we call intra-molecular nonadiabatic electronic energy redistribution (INEER) due to hyper-resonance [188]. This makes a contrast to intramolecular vibrational energy redistribution (IVR) caused by classical chaos on a single potential basin. INEER is a crucial consequence of the intense chaos due to the continual nonadiabatic electronic-state mixing.

## 8. Concluding Remarks

To review “quantum chaos” in the scope of chemical dynamics, we herein have tracked the following subjects. We began with full quantum dynamics, wavepackets and eigenfunctions localized in the quasi-separatrix in two-dimensional potential function. This study was made in search for a novel molecular vibrational mode. Indeed, in this small dimensional system lie several vibrational modes, which give birth to very long-time wandering motions in the quasi-separatrix. A pair of the eigenstates of the dynamical tunneling of the second kind have been identified.

Semiclassical theory and energy quantization thereof, which is indispensable in the study of molecular vibration were next reviewed. We took a wavepacket approach in studying semiclassical dynamics rather than the direct semiclassical reduction of the Feynman kernel. With this approach, we discussed the mechanism of energy quantization for not only making the peaks but suppressing the off-resonant components. One of the most annoying parts in semiclassical kernel and related quantities is the divergence at caustics and/or turning points, which casts an inherent difficulty to the Gutzwiller trace formula. To avoid the difficulty, we have shown that the amplitude-free energy quantization actually works quite well. Chaotic paths that appear only through quantum tunneling have been shown along with a semiclassical tunneling formalism based on the notion of the parity of motion.

One of the main aims of the present review has been to guide the readers to chaos in molecular nonadiabatic dynamics, which is induced by the kinematic couplings between nuclei and electrons. The entanglement between electrons and nuclei is one of the most important subjects in general chemical science. We have emphasized the critical roles of “bifurcation and merging” of quantum wavefunctions. The most characteristic molecular chaos is found in nonadiabatic electron dynamics in the densely quasi-degenerate electronic-state manifold, every state of which mixes together with other members of the manifold. We have shown that the dynamical properties of electronic-state chaos can be measured in terms of the quantities related to diffusion and memory loss in the Hilbert space, turbulent flow of electron current, and so on. The chaos of the electronic excited state should become more and more vital in future molecular science, which will proceed far beyond the Born-Oppenheimer paradigm of the current theoretical framework of chemistry.

Some of the characteristics found for molecular quantum chaos are summarized below:

The basic mechanism of chaos in nonadiabatic wavepacket dynamics is “bifurcation and merging”, which is a quantum mechanical synonym of “stretching and folding” in the classical baker’s (or the Smale horse-shoe) transformation. Indeed repeated occurrences of “bifurcation and merging” of quantum wavepackets result in genuine chaos in quantum mechanics not only in energy spectra but more importantly in the dynamics of wavepackets. We would like to particularly emphasize again that the bifurcation and merging of wavepackets are indeed physically observable. Interestingly, “bifurcation (branching) and merging of the reaction tubes” sets also a geometrical foundation of classical chaos in structural isomerization dynamics. Extension of the tube geometry to quantum mechanics has not yet been explored.

“Loss of memory” is one of the central concepts to characterize chaos in general. The positive Lyapunov exponent in classical systems indicates an exponential separation of nearby orbits, leading to the loss of the memory of their initial stage. In our quantum chaos, the exponential decay of the memory of an electronic state in the time-propagation against the nonadiabatic (kinematic) interactions can take place in the Hilbert space. Again, we have observed that classical chaos in the structural isomerization dynamics of liquid-like clusters is associated with a characteristic feature of memory loss, which leads to a random yet statistical appearance of the isomers in the time series.

The advent of the ultrafast pulse laser has revolutionized molecular science since the 1970s. The femtosecond laser chemistry [189] to date has made it possible to real-time track the molecular deformation and reactions, an example of which can be seen in Figure 11 of this review. We are now in the age of technical development and applications of attosecond pulse lasers, the width of which is compatible with the time scales of electronic motions in molecules. This situation has been driving us in nonadiabatic electron wavepacket dynamics in Section 7. Nevertheless, we also call for attention for the study of extremely long-time spectroscopy. As we noted in this article, the structural isomerization of not only clusters but also large molecules like proteins can wander from one shape (potential valley, potential basin) to another taking a very long time at each local structure. These many-dimensional dynamics are often accompanied by large dimensional chaotic spaces, which makes the entire dynamics even longer and more complicated. It is extremely challenging therefore to figure out spectroscopic means to capture the essential feature of the long-time dynamics. The time-dependent spectroscopy, which was proposed in Section 4 and discussed briefly in the section of semiclassical mechanics Section 5, is just one primitive idea in an attempt to cope with this matter. A parallel computation scheme with many different initial conditions should be a good candidate to overcome long-time quantum dynamics in ergodic phase space, if a method to appropriately connect the pieces of spectra taken individually is devised.

A final remark before closing. The essential nature of classical and quantum chaos may lie in mathematical beauty and intricate geometry behind the complicated-looking phenomena. However, the complexity of molecules often makes such elegant mathematical structures obscure and even less mighty. Moreover, We are afraid that sticking to the too rigorous definitions of quantum chaos would isolate itself from other natural science. Therefore, I would like to conclude this review by stressing that more studies on quantum chaos made mathematically easy and conceptually useful are demanded than ever, since there are so many unknowns left (not only) in the molecular phenomena, notions, and laws that need an understanding of quantum chaos.

## Figures and Tables

**Figure 1 entropy-25-00063-f001:**
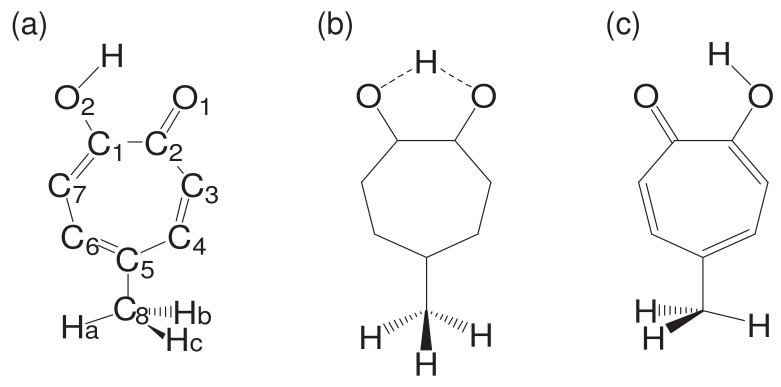
(**a**) Stable structure of 5-methyltropolone (5MTR). Hydrogen atoms attached on C${}_{3}$, C${}_{4}$, C${}_{6}$, and C${}_{7}$ are not shown. (**b**) The transition state on the way. (**c**) Proton has ben shifted from O${}_{2}$ to O${}_{1}$ inducing the internal rotation of the methyl group at the bottom. [Taken from Ushiyama, H.; Takatsuka, K. *Angew. Chem. Int. Ed.* **2005,** *44*, 2 with permission].

**Figure 2 entropy-25-00063-f002:**
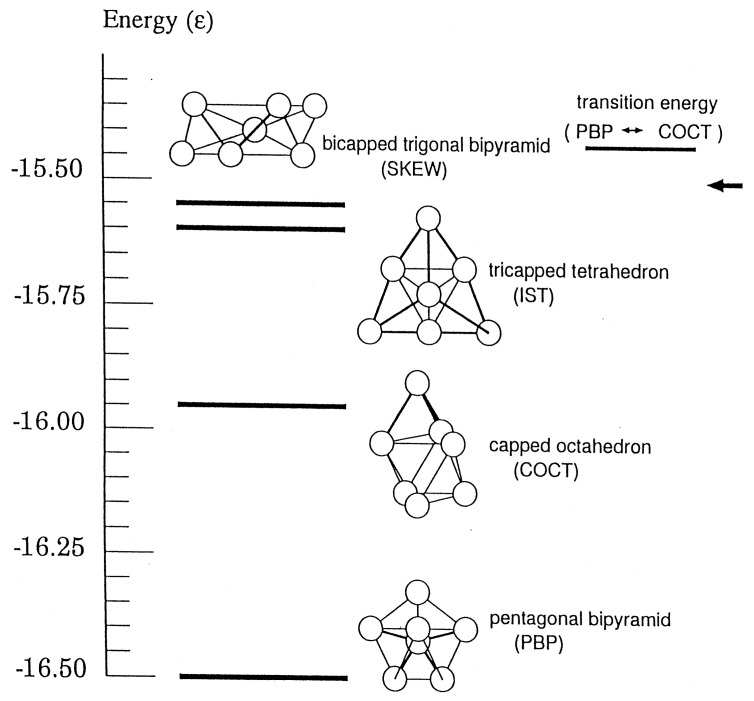
Four isomers on the single Lennard-Jones potential energy surface. The names and the bottom energy of each potential basins are presented. [Drawn by Dr. C. Seko].

**Figure 3 entropy-25-00063-f003:**
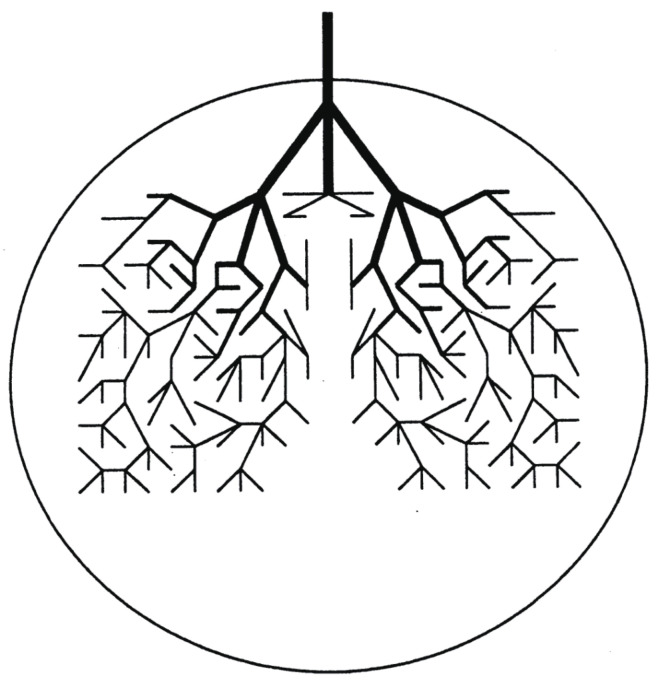
Schematic presentation of a unit of reaction tube in a potential basin and its bifuraction structure. The circle schematically models a potential basin, each of which supports one isomer. Only one of tubes is depicted and truncated at some points. The reaction tubes lie behind classical chaos in structural isomerization dynamics, which is one of the so-called many-valey problems. The similar tube units are extended from every neighboring potential basins, and one basin is filled with many tubes. Time-revesal symmetry gives an idea of how the reaction tubes are reconnected with one another to form thicker tubes and get out of the basin to next ones. [Drawn by Dr. C. Seko].

**Figure 4 entropy-25-00063-f004:**
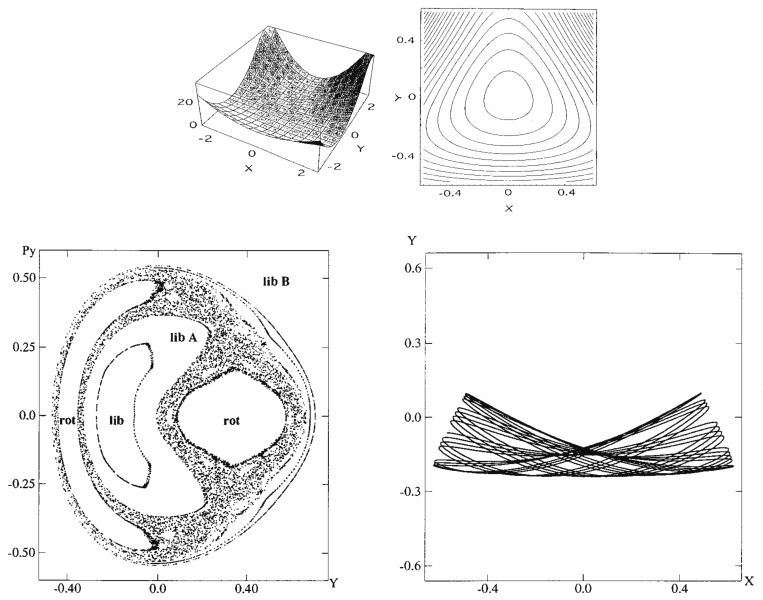
Panels in the top row for the potential functions used (the left one for a perspective view and the right for the contour plot). The bottom row, left panel indicates the classical Poincaré surface of section at x=0 and $p_{x}>0.$ The dark area with many dotts displays the quasi separatorix separating the major tori. The symbols “lib” and “rot” characterize the vibrational modes of the individual tori. The bottom right panel exemplifies one of the torus modes, a libratation. [Taken from Hashimoto, N.; Takatsuka, K. *J. Chem. Phys*. **1998,** *108*, 1893 with permission].

**Figure 5 entropy-25-00063-f005:**
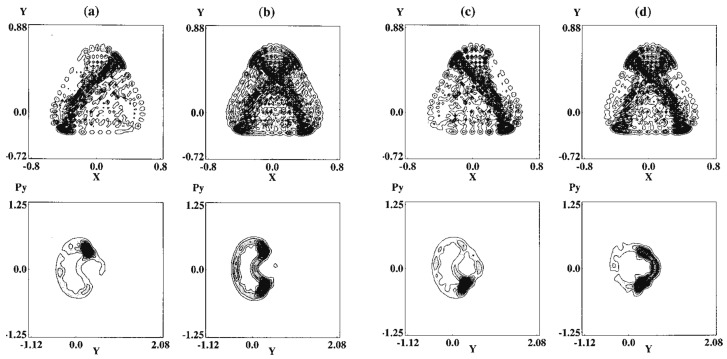
Time propagation of a quantum wavepacket in the quasi-separatrix. Time passes from panel (**a**–**d**). The upperrow indicates ${\left|\psi \left(x,y,t\right)\right|}^{2}$ at selected times, while the lower row shows the corresponding quantum Poincaré section, in which the wavepacket is seen to be running on the classical quasi-separatrix. [Taken from Hashimoto, N.; Takatsuka, K. *J. Chem. Phys*. **1998,** *108*, 1893 with permission].

**Figure 6 entropy-25-00063-f006:**
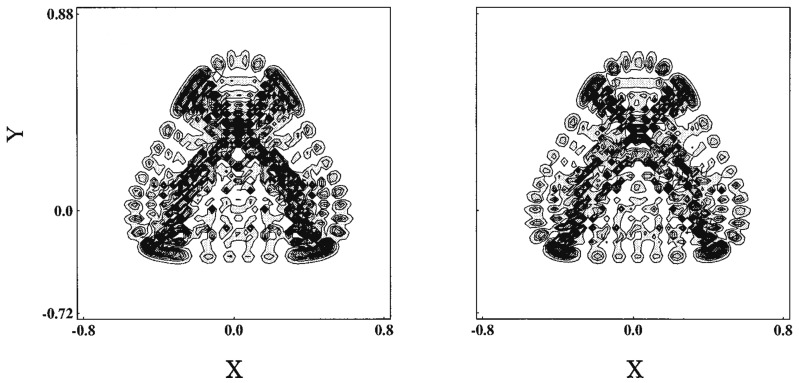
A pair of almost degenerate eigenfunctions lying in the quasi-separatrix. In the left panel is an odd function with respect to the symmetry axis x=0 [energy E=0.152094], while the right being even [E=0.152500]. The slight energy difference and the separation of the symmetry are due to what we call dynamical tunneling of the second kind. See the text. [Taken from Hashimoto, N.; Takatsuka, K. *J. Chem. Phys*. **1998,** *108*, 1893 with permission].

**Figure 7 entropy-25-00063-f007:**
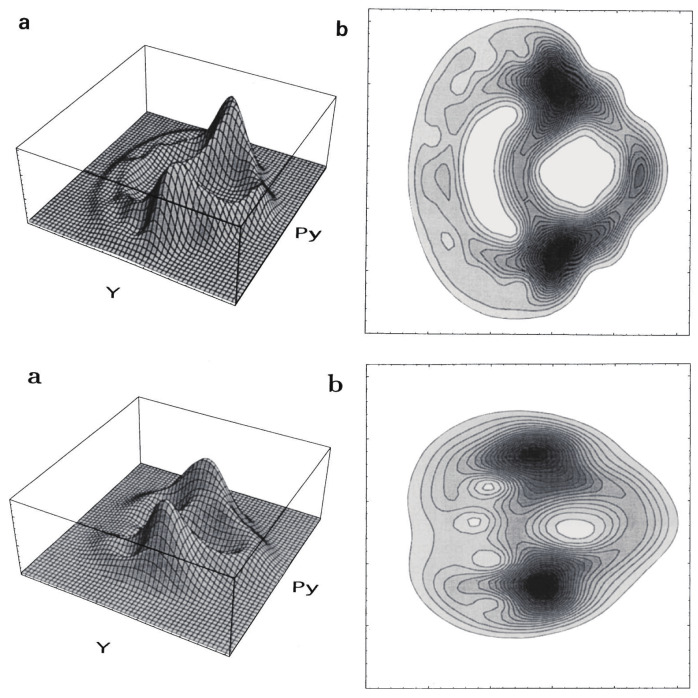
Quantum Poincaré surface of section. Panels at the top row reprsent an quasi-separatorix eigenfunction given by ℏ=0.005 ($\left[E_{1},E_{2}\right]=\left[0.150.0.153\right]$). The bottom panels are the eigenfunction at ℏ=0.010 ($\left[E_{1},E_{2}\right]=\left[0.150.0.156\right]$). Left column (**a**) for the perspective views and the right (**b**) for contour plots. [Taken from Hashimoto, N.; Takatsuka, K. *J. Chem. Phys*. **1995,** *103*, 6914 with permission].

**Figure 8 entropy-25-00063-f008:**
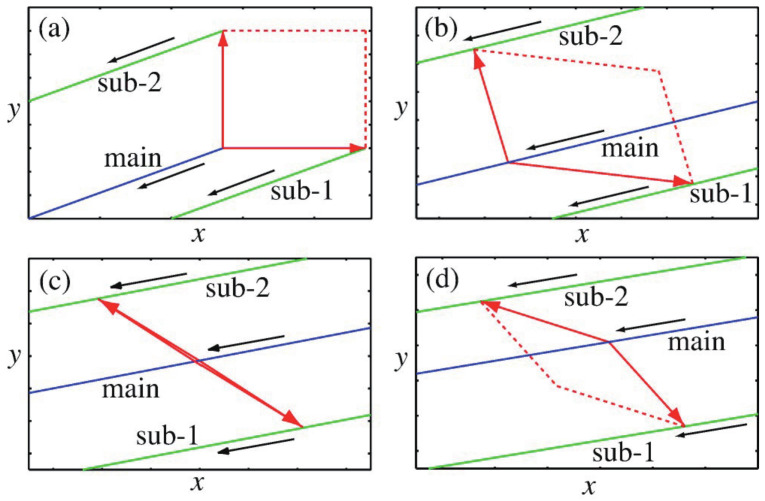
An illustration for geometrical implication of $\sigma \left(t\right)$ defined in Equation (Equation 36). A deviation determinant $\sigma \left(t\right)$, which is an orientable volume, withrespect to the main path is formed with the help of nearby classical paths. Time passes from panel (**a**–**d**). [Taken from Takahashi, S.; Takatsuka, K. *Phys. Rev. A* **2004,** *69*, 022110 with permission].

**Figure 9 entropy-25-00063-f009:**
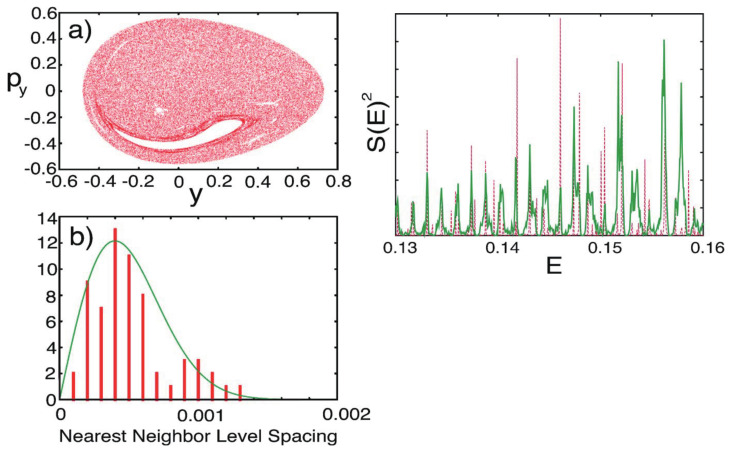
(**a**) Classical Poincaré surface of section at x=0, $p_{x}=0$ for an ensemble of trajectories with the total energy $E_{cl}$=0.157. ℏ=0.005. (**b**) The nearest neighbor level spacing of the full quantum spectrum in this energy region, which is subject to the Wigner distribution, an indication of quantum chaos. The right panel shows the spectrum taken from the amplitude free correlation function (green solid) and full quantum counterpart (red). Only the spectral positions should be compared. [Taken from Takahashi, S.; Takatsuka, K. *J. Chem. Phys.* **2007**, *127*, 084112 with permission].

**Figure 10 entropy-25-00063-f010:**
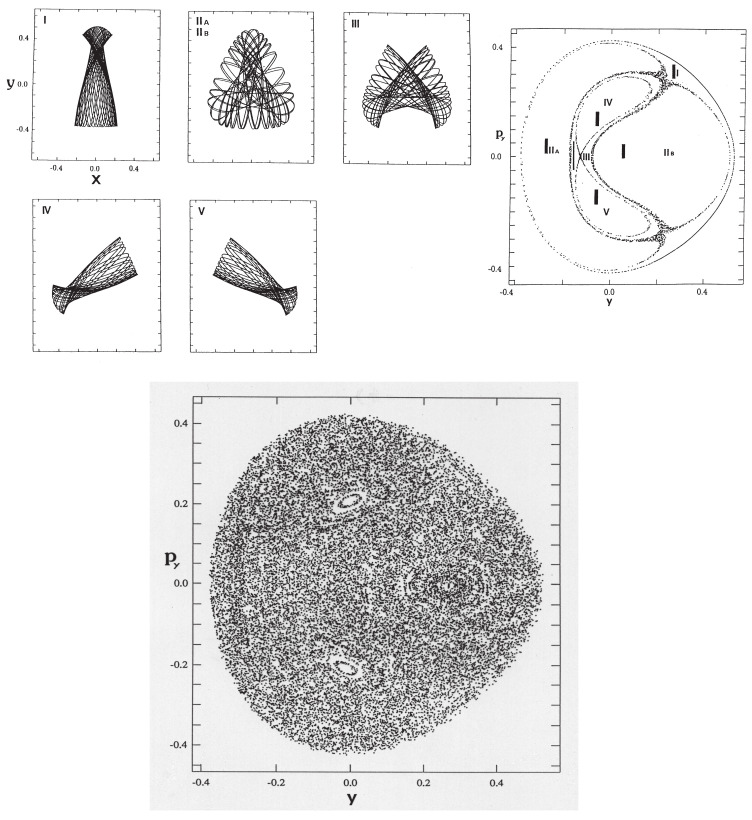
The right panel of the upper row shows the Poincaré section (x=0 and $p_{x}>0$) along with the vibrational modes (left) belonging to the tori. E=0.09. The botton row shows a long time plot at the Poincaré section, which wanders from one torus to another through tunneling. Statistical distribution through tunneling is recognized. [Taken from Ushiyama, H.; Takatsuka, K. *Phys. Rev. E* **1996,** *53*, 115 with permission].

**Figure 11 entropy-25-00063-f011:**
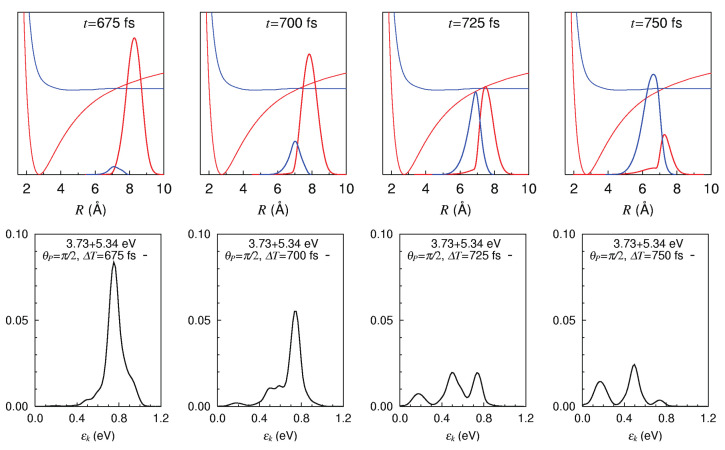
Photoelectron kinetic energy spectrum as the wavepackets bifurcate and merge at the avoided crossing. (The potential curves in this figure are graphcally represented in the *diabatic* representation.) The polarization vector of the probe laser light is perpendicular to that of the pump, while the latter is set parallel to the line between Na and I. In the upper row are shown the absolute square of the wavepackets at selected times (red (blue) one on the red (blue) potential curve). The bottom row represents the corresponding energy-resolved photelectron signals. Figure drawn by Dr. Y. Arasaki. [Taken from Takatsuka, K. *Bull. Chem. Soc. Jpn.* **2021**, *94*, 1421 with permission].

**Figure 12 entropy-25-00063-f012:**
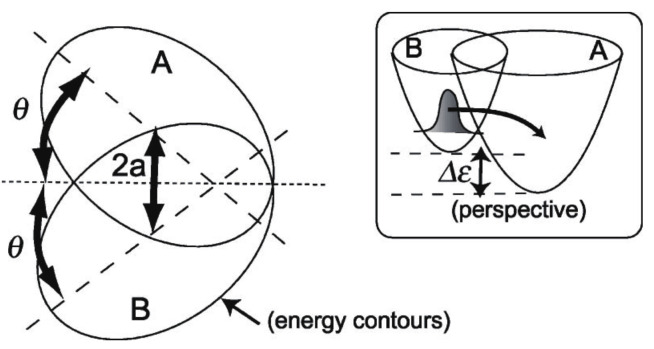
A schematic representation of Heller’s model system. The distance between the centers of the potentials is 2a, and the angle between the relevant crossing seam (dotted line) and the primary axis of each potential (dashed line) is θ. Inset: The perspective view of the present system. The energy difference of the potential bottoms is $\Delta \epsilon =\epsilon_{B}-\epsilon_{A}$. [Taken from Fujisaki, H.; Takatsuka, K. *J. Chem. Phys*. **2001,** *114*, 3497 with permission].

**Figure 13 entropy-25-00063-f013:**
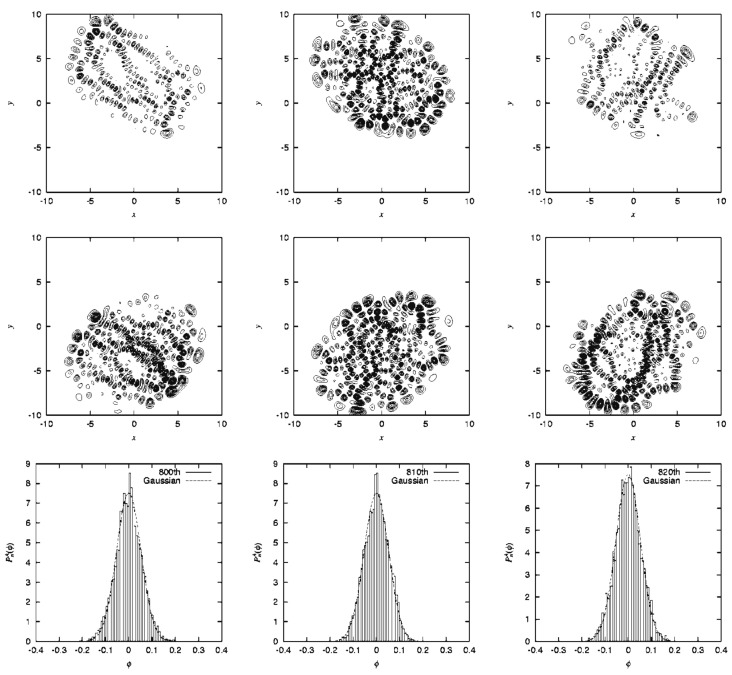
First row from the top: Eigenfunctions squared (from left to right, 800th, 810th, and 820th) of Heller’s model system on surface *A* (see Figure 12) with J=1.5 (intermediate coupling) and θ=π/6. Second row: The corresponding eigenfunctions (squared) on surface *B*. Third row: The corresponding amplitude distributions of the eigenfunctions onsurface *A*. The dashed envelope curves represent Gaussian functions. [Taken from Fujisaki, H.; Takatsuka, K. *Phys. Rev. E* **2001,** *63*, 066221 with permission].

**Figure 14 entropy-25-00063-f014:**
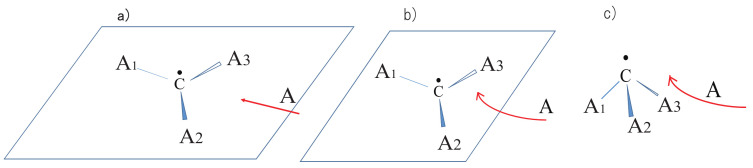
A symmetry breaking by pyramidalization of a coplanar molecular system, leading to molecular chirality. (**a**) Atom A is approaching the planar molecule on the same plane. (**b**) The Lorentz-like nonadiabatic force triggers to guide the incident atom to the direction out of the plane. (**c**) The Hellmann-Feynman force follows to make the pyramidalization complete. [Taken from K. Takatsuka, J. Chem. Phys. **146**, 084312 (2017) with permission].

**Figure 15 entropy-25-00063-f015:**
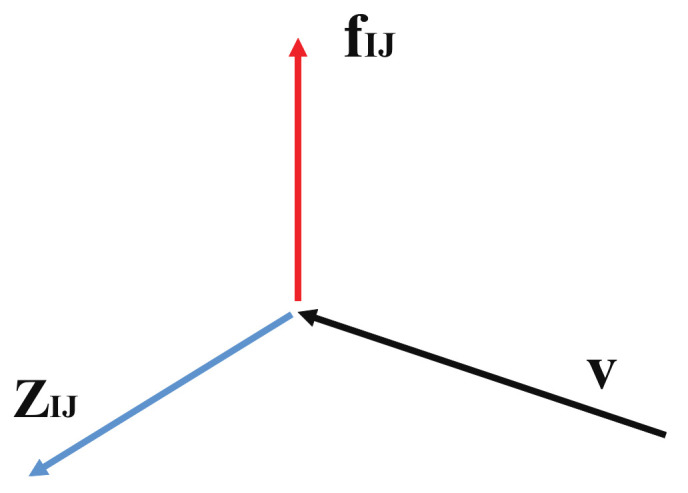
A particle coming in with a velocity vector v into the $Z_{IJ}$ field feels a force $f_{IJ}$, which is orthogonal to both v and $Z_{IJ}$, according to ${\vec{f}}_{IJ}^{a}\left(R\right)={\vec{v}}^{a}\times {\vec{Z}}_{IJ}^{{}_{a}}\left(R\right).$ See the text for the definition of $Z_{IJ}$.

**Figure 16 entropy-25-00063-f016:**
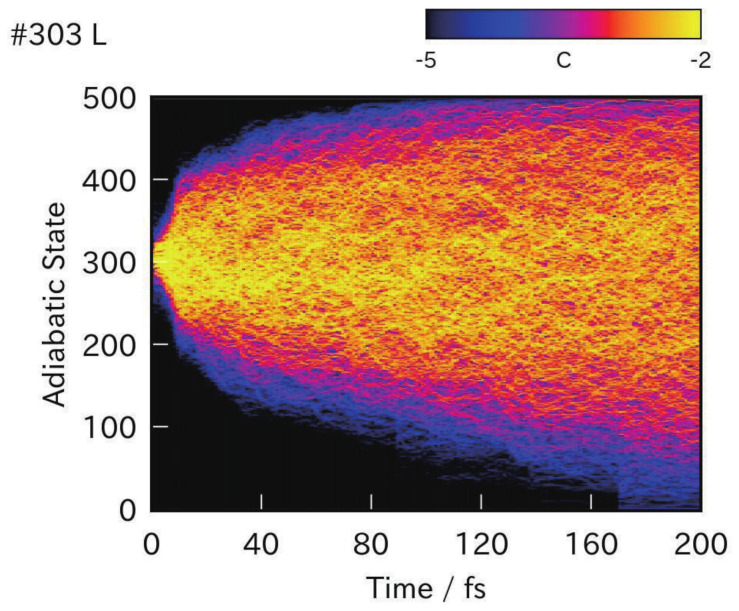
Diffusion–like redistribution of the electronic states in the excited electronic state manifold of B${}_{12}$ cluster. It is actually a fractional Brownian motion [164] in the state space. The dynamics of the log${\left|C_{I}\left(t\right)\right|}^{2}$ of Equation (Equation 108) is tracked with the SET and is charted in color code. An initial state has been prepared at the 300th adiabatic state. The extensive diffusive state is realized well before 10 fs. [Figure drawn by Dr. Y. Arasaki. Taken from Takatsuka, K. *Bull. Chem. Soc. Jpn.***2021***, 94*, 1421 with permission].

**Figure 17 entropy-25-00063-f017:**
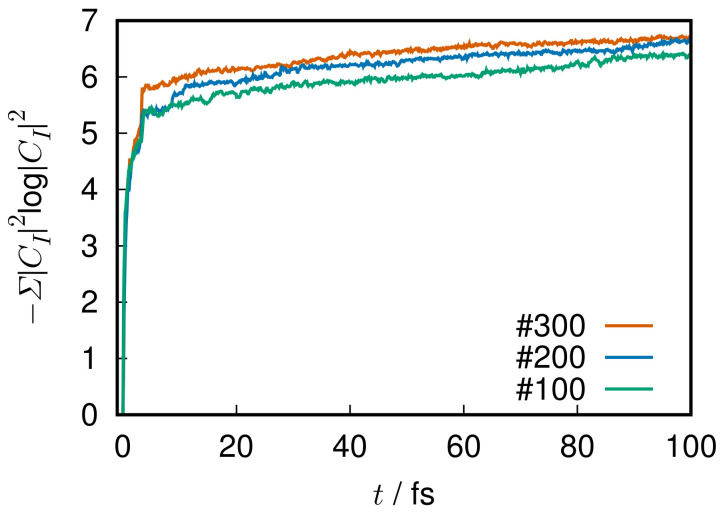
Time evolution of the Shannon entropy with respect to the adiabatic state population ${\left|C_{I}\left(t\right)\right|}^{2}$ involved in the wavepackets, each starting from the adiabatic state #100, #200, and #300, respectively. [Provided by Dr. Yasuki Arasaki].

**Figure 18 entropy-25-00063-f018:**
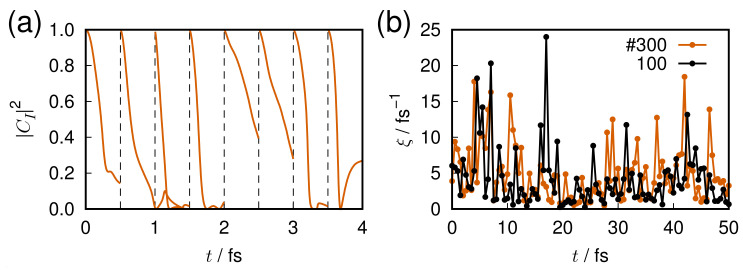
Loss of memory of nonadiabatic states in the Hilbert space. (**a**) Quick decay of ${\left|C_{I}\left(t\right)\right|}^{2}$ from 1.0 to $C_{I}\left(t+\Delta t\right)$ with Δt=0.5 fs at selcted points of *t* for B${}_{12}$ cluster SET dynamics starting from the 300th adiabatic state. Dynamics starting at t=0.0, 0.5, 1.0, 1.5, 2.0, 2.5, 3.0, and 3.5 fs. (**b**) Exponent of memory loss, defined in Equation (Equation 128), for the initial adiabatic component weight, ξ(t,Δt), starting from either the 300th (red) or the 100th (black) adiabatic state. Δt=0.5 fs. Lines connect calculated points only for visualization. [Provided by Dr. Yasuki Arasaki].

**Figure 19 entropy-25-00063-f019:**
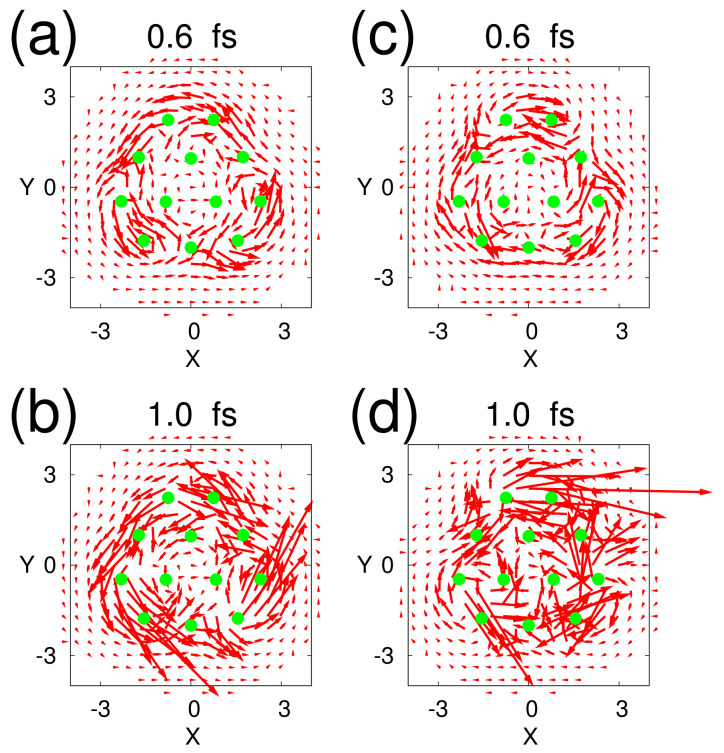
Spontaneous electron flux arising from the wavepackets in the SET dynamics starting from the 74th adiabatic state at (**a**) 0.6 fs and (**b**) 1.0 fs, and from the 75th adiabatic states at (**c**) 0.6 fs and (**d**) 1.0 fs. dynamics. The flux vectors are systematically multiplied by $2.0\times 10^{4}$ for visualization. These fluxes are driven by nonadiabatic interactions. Soon after 1.0 fs, the regular-looking fluxes become turbulent. [Taken from Yonehara, T; Takatsuka, K. *J. Chem. Phys.***2016***, 144,* 164304 with permission].

**Figure 20 entropy-25-00063-f020:**
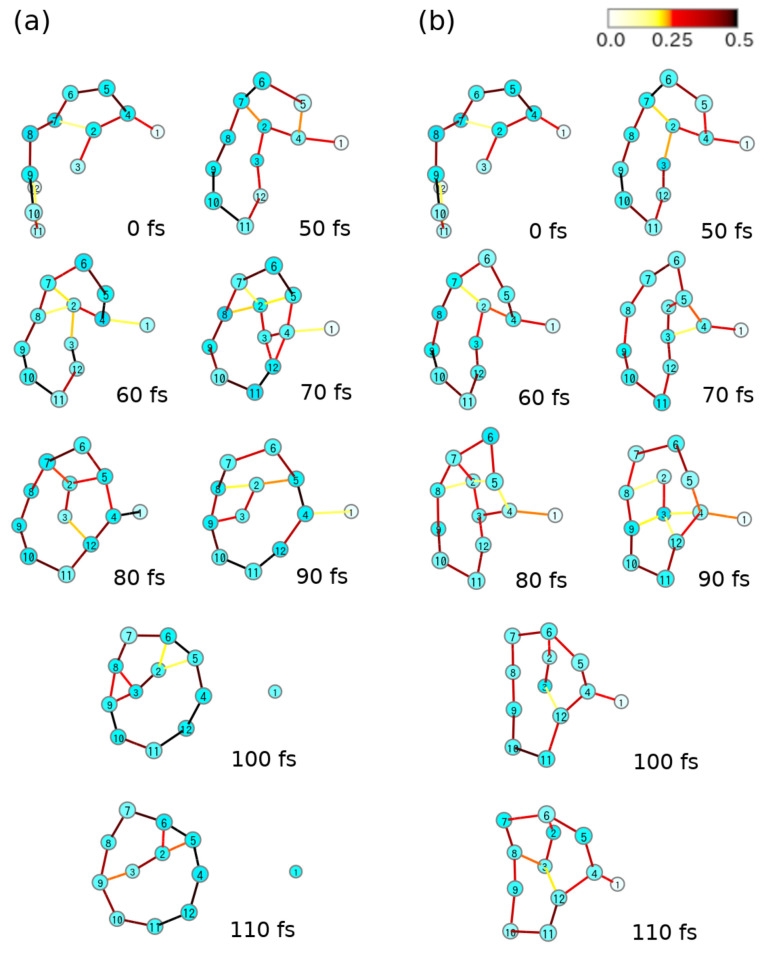
B${}_{12}$ bonding structures along a trajectory under (**a**) adiabatic dynamics, and (**b**) nonadiabatic dynamics, starting from the same initial conditions (initial adiabatic state 47). Time *t* in fs is indicated beside each structure. At t=110 fs, atom 1 is in the dissociation channel in the adiabatic dynamics, while this atom is pulled back to the cluster site in the nonadiabatic dynamics. [Taken from Arasaki, Y.; Takatsuka, K. *J. Chem. Phys.* **2019,**
*150*, 114101 with permission].

## Data Availability

All data included in this manuscript are available upon reasonable request by contacting the corresponding author.

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
