# Peer review of "Quantum Chaos in the Dynamics of Molecules"

_entropy, 2022, doi:10.3390/e25010063_

Round 1

Reviewer 1 Report

This is a review of quantum and semiclassical dynamics of molecular systems with reference to chaotic features of the time-dependent dynamics and eigenvalue spectra. Quantum chaos in the vibrational spectra of anharmonic few-dimensional models is addressed. Semiclassical nuclear wave-packet dynamics including tunnelling is outlined. The main emphasis of the article is on quasi-classical nonadiabatic electronic-nuclear wave-packet dynamics. The connection of these phenomena, such as wave-packet bifurcation and merging, to quantum chaos is more intuitive than rigorous, since true quantum chaos cannot be identified on femtosecond or picosecond time scales. More rigorous concepts of quantum chaos in strongly nonadiabatically coupled molecular systems, such as eigenvalue spacing distributions, are not covered. Although nonadiabatic dynamics is a major focus of the review, the concept of conical intersection, which certainly is at the very core of nonadiabatic dynamics in polyatomic molecules, is not mentioned on 46 pages of review, which is amazing. The viewpoint of the author is rather personal and the references are strongly biased toward the work of the author’s group. Nevertheless, this article is a useful summary of the extensive work of the Takatsuka group over the past 30 years. The meaning of “review from the backyard” in the title is not clear to this reviewer. The article is well written, but still contains many typos.

Reviewer 2 Report

The manuscript presents a review of quantum chaos of molecular systems. The author reviews the Born-Oppenheimer approximation, essential for topics discussed later; implications of chaos to molecular science, in particular chemical reactions; quantum chaos of a 2-dimensional system; a semiclassical method for studying dynamics of nuclear motion in molecules, which was developed by the author; then quantum chaos of non-adiabatic electron dynamics. The latter, reviewed in section 7, is a field pioneered and developed by the author, and is a major focus of the review. Overall, this is a beautiful presentation of a large body of work that has had and will continue to have a large impact on molecular science.

The impact of this valuable and important review might be greater if some additional aspects of chaos in molecular systems were brought out. The author might consider mentioning these points:

1. On page 2, the author points out studies of chaos in low-dimensional molecular systems, typically two dimensional. However, there were classical studies already in the 1980s that considered classical chaos in larger molecular systems and atomic clusters (also discussed in the review article), including “Melting and phase space transitions in small clusters: spectral characteristics, dimensions, and K-entropy,” JCP 89 1681 - 1694 (1988). There is also more recent work by Keshavamurthy with comparison to quantum dynamics (for example, “Relevance of the Resonance Junctions on the Arnold Web to Dynamical Tunneling and Eigenstate Delocalization,” J. Phys. Chem. A 122, 8636 - 8649 (2018). These might be mentioned.

2. Section 3 discusses studies of chaos towards a better understanding of chemical dynamics. Here the author might point out that transitions to both classical and quantum chaos have explained kinetics of chemical reactions involving large molecules. Chandler discussed the role of the onset of classical chaos in the isomerization of cyclohexane (“Stochastic MD simulation of cyclohexane isomerization,” J. Phys. Chem. 92, 3261 - 3267 (1988)) and the role of quantum ergodicity and chaos on cyclohexane isomerization has been addressed as well (“Heat transport in molecules and reaction kinetics: The role of quantum energy flow and localization,” Adv. Chem. Phys. 130B, 205 - 256 (2005)), as has the role of the onset of quantum ergodicity and chaos in the isomerization of other organic molecules in the ground electronic state (“Influence of quantum energy flow and localization on molecular isomerization in gas and condensed phases,” Int. J. Quant. Chem 75, 523 - 531 (1999)). The role of the onset of quantum ergodicity and chaos on isomerization of molecules in excited electronic states has also been examined (“Quantum energy flow during molecular isomerization,” Chem. Phys. Lett. 280, 411 - 418 (1996)). The study of reactive molecular systems from classical/semiclassical and quantum mechanical points of view using ideas from chaos theory continues to be an active research field (for example, “Stable chaos and delayed onset of statisticality in unimolecular dissociation reactions,” Comm. Chem. 3, art. no. 4, (2020)).

3. On page 6 there is a discussion of the relevance of the total density of states of molecules to stochastic dynamics. While the total density of states is important, the onset of stochasticity and quantum ergodicity/chaos is directly related to the local density of resonantly coupled states. This is what controls classical and quantum chaos in many-dimensional systems such as molecules. The notion of intersection of quantum resonances towards the onset of quantum ergodicity and chaos was discovered by Logan and Wolynes (“Quantum localization and energy flow in many-dimensional Fermi resonant systems,” JCP 93, 4994 - 5012 (1990)), and further developed since then (for example, “Vibrational relaxation and energy localization in polyatomics:  Effects of high-order resonances on flow rates and the quantum ergodicity transition,” JCP 105, 11226-11236 (1996)). Important reviews of this work and its impact on molecular science, which might be mentioned, include “Quantum ergodicity and energy flow in molecules,” Adv. Phys. 64, 445 - 517 (2015), and “Intramolecular vibrational energy redistribution and the quantum ergodicity transition: a phase space perspective,” PCCP 22, 11139 - 11173 (2020). The notion of intersection of resonances in the onset of quantum chaos in molecules has its analogy in the Chirikov resonance overlap criterion for classical chaos. 

Some other comments:

4. Figure 9(b), In the caption, the author states, “…which is subject to the Wigner distribution”. Perhaps it would be worthwhile adding something like, “…Wigner distribution, an indication of quantum chaos.” The reason is that the author discusses the implication of the Wigner distribution only later in the article, in section 6, so that the reader may not recognize yet the meaning of the comparison that is made in the figure.

5. In the abstract, the author refers to “Intermolecular Nonadiabatic Electronic Energy Redistribution”. Is Intramolecular meant in this case, as this is what is discussed later on?
